# Unraveling the orientation of phosphors doped in organic semiconducting layers

Chang-Ki Moon[1], Kwon-Hyeon Kim [1] & Jang-Joo Kim[1]

Emitting dipole orientation is an important issue of emitting materials in organic light-emitting diodes for an increase of outcoupling efficiency of light. The origin of preferred orientation of emitting dipole of iridium-based heteroleptic phosphorescent dyes doped in organic layers is revealed by simulation of vacuum deposition using molecular dynamics along with quantum mechanical characterization of the phosphors. Consideration of both the electronic transitions in a molecular frame and the orientation of the molecules at the vacuum/molecular film interface allows quantitative analyses of the emitting dipole orientation depending on host molecules and dopant structures. Interactions between the phosphor and nearest host molecules on the surface, minimizing the non-bonded van der Waals and electrostatic interaction energies determines the molecular alignment during the vacuum deposition. Parallel alignment of the main cyclometalating ligands in the molecular complex due to host interactions rather than the ancillary ligand orienting to vacuum leads to the horizontal emitting dipole orientation.

---

[1] Department of Materials Science and Engineering, RIAM, Seoul National University, Seoul 151-744, South Korea. Correspondence and requests for materials should be addressed to J.-J.K. (email: jjkim@snu.ac.kr)

The orientation of molecules in molecular films dictates their electrical and optical properties such as charge mobility[1, 2], birefringence[3], absorption[4], emission[5], ionization potential[6], and dielectric[7] and ferroelectric properties[8]. Therefore, understanding and control of molecular orientation in organic films have been a research topic with central importance in organic electronics and photonics, including the fields of liquid crystals[9], organic field effect transistors[10], and organic photovoltaics[11]. In organic light-emitting diodes, the molecular orientation of emitter embedded in the emissive layer has been an issue to enhance the outcoupling efficiency of light pursuing the horizontal alignment of the emitting dipole moment[3, 12–21].

Interestingly enough, it is only in recent years has attention turned to the orientation of emitting dipoles of iridium-based phosphors, the most verified light-emitting dyes with high photoluminescence quantum yield and variety of chromatic spectrum as doped in the emissive layers; probably because their iridium-centered spherical shape and the amorphous surrounding nature in the emissive layers are far from having strong molecular alignments. Recently, some heteroleptic Ir complexes exhibiting efficient electroluminescence in organic light-emitting diodes are reported to possess preferred horizontal emitting dipole orientations (EDOs)[13–16, 18–20]. However, it was difficult to assert the reason why the spherical-shaped phosphors have a propensity toward preferred molecular alignment in the emissive layers. A few mechanisms have been proposed to explain the preferred molecular orientation of the Ir complexes doped in vacuum-deposited organic semiconducting layers: molecular aggregation of the dopants leading to randomizing their orientation by suppressing the intermolecular interaction between the dopant and host molecules[22], strong intermolecular interactions between electro-positive sides of the dopant, and the electro-negative host molecules promoting parallel alignment of the N-heterocycles of Ir complexes by forming host-dopant–host pseudo-complex mainly participating in $^3$MLCT transition[16, 23], and π–π interactions between the dopant and host molecules on the organic surface bringing alignment of aliphatic ligands to the vacuum side[20, 24]. Currently, it is not very clear which mechanism most comprehensively describes the origin of the preferred EDO of the heteroleptic iridium phosphors. Moreover, the models are too oversimplified to describe the EDO values quantitatively, which depend on structures of the phosphors and host molecules[23]. Therefore, the molecular configurations and the interactions responsible for the EDOs of Ir complexes should be established by atomic-scale simulation of the Ir complexes interacting with host molecules during film fabrication.

In this paper, we carefully examine the vacuum deposition process of phosphors on organic layers using a combination of

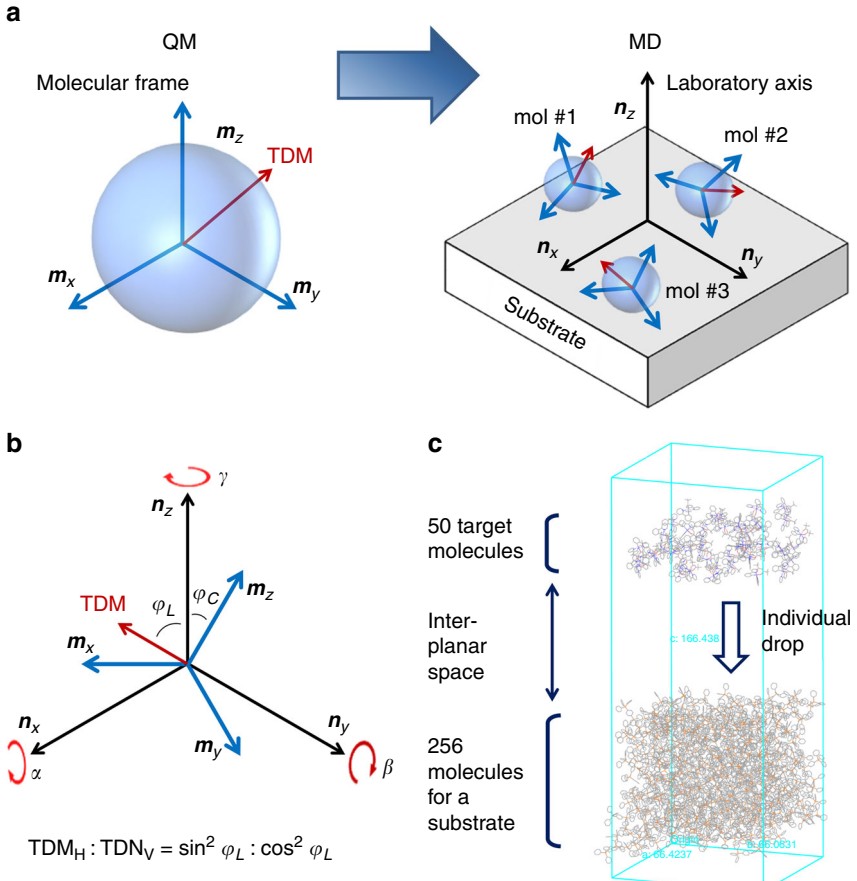

**Fig. 1** Method for the simulation of the EDO of emitters in vacuum-deposited layers. **a** Transfer of the TDM vectors (red arrow) in the molecular coordinates to the vectors of the molecules on the organic substrate during the vacuum deposition simulation. **b** Three rotation angles (α, β, and γ for the clockwise rotation to the $n_x$-, $n_y$-, and $n_z$-axes, respectively) were the orientation parameters of the molecules to correlate the molecular orientation to the laboratory axis. Angles between $n_z$ axis and the TDM vector ($\varphi_L$) and the $C_2$ axis ($\varphi_C$) are obtained after the vector transformation. **c** A simulation box consisting of the substrate and target molecules. About 50 target molecules were located above the substrate with 5.0 nm of inter-planar space dropped individually at 300 K. The distance unit in the figure is angstrom (Å)

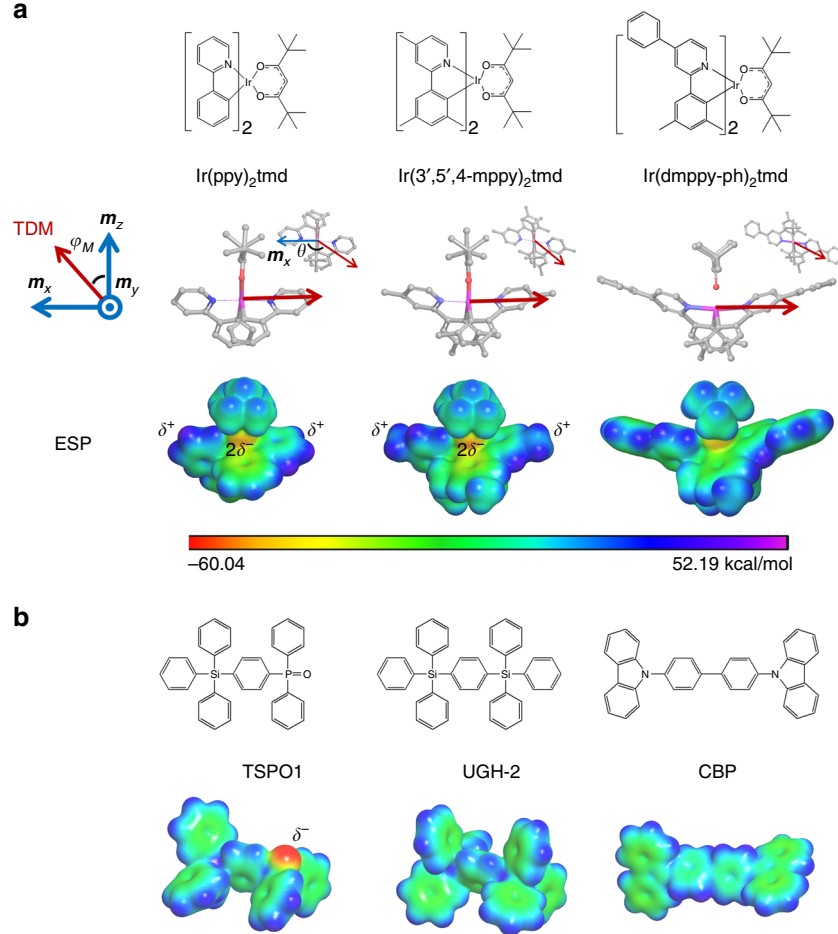

**Fig. 2** Iridium complexes and host materials. **a** Chemical structures, transition dipole moment vectors, and electrostatic potentials of Ir(ppy)$_2$tmd, Ir(3′,5′,4-mppy)$_2$tmd, and Ir(dmppy-ph)$_2$tmd phosphors. There are linear quadrupoles in the ground state of Ir(ppy)$_2$tmd and Ir(3′,5′,4-mppy)$_2$tmd with quadrupole moments along the principal axes of $Q_{xx,yy,zz}$ = [25.2,−13.0,−12.2] and [27.1,−12.8,−14.3] Debye·Å$^2$, respectively. **b** Chemical structures and electrostatic potentials of UGH-2, CBP, and TSPO1 host molecules. The electrostatic potentials are projected on the isosurface of electron density of 0.005 electrons/bohr$^3$. The *color legend* is identical to that for Ir complexes. Optimization of the molecular structures was demonstrated using B3LYP method and LACVP** basis set for the phosphors and 6–31 g(d)** for the host materials, respectively. SOC-TDDFT of the phosphors were carried out using B3LYP method and DYALL-2ZCVP_ZORA-J-PT-GEN basis set

molecular dynamics (MD) simulations and quantum mechanical analyses. The triplet EDO of heteroleptic Ir complexes doped in organic layers is studied with systematic variations of the molecular structures of both host and dopant. Theoretical prediction of EDO from simulated deposition process reveals excellent quantitative agreement with experimental observations, reproducing the anisotropic molecular orientations of heteroleptic Ir complexes in the emissive layers. In-depth analysis indicates that the molecular orientation originates from the coupling of the cyclometalated main ligand participating in the optical transition with neighbor host molecules rather than from the alignment of aliphatic ancillary ligand toward the vacuum. Close observation of the simulation results indicates that non-bonded interaction energy has a critical influence on the molecular orientation during the deposition.

## Results

**Modeling of emitting dipole orientation.** The simulation method for obtaining EDO of an emitter in the vacuum-deposited layer is schematically illustrated in Fig. 1a. First, the transition dipole moment (TDM) vector in the molecular frame ($m_x$-, $m_y$-, and $m_z$-axes) was determined by quantum mechanical

calculations after optimization of molecular geometry. For iridium-based phosphors, spin-orbit-coupled time-dependent density functional theory (SOC-TDDFT) was employed for the calculation of the triplet TDM vectors for phosphorescence. Second, vacuum deposition of the emitting molecules on organic surfaces was simulated using MD. Finally, the TDM vectors in the molecular axis in each frame of MD were transformed to the vectors in the laboratory axis ($n_x$-, $n_y$-, and $n_z$-axes) by rotation matrix method (Fig. 1b). We determine $\varphi_C$ and $\varphi_L$ as the angle between $m_z$ and $n_z$ axes and the angle between the TDM vector of the emitter and $n_z$ axis, representing the molecular orientation and the EDO against the vertical direction in the laboratory axis, respectively. The ratio of the horizontal (TDM$_H$) to the vertical transition dipole moment (TDM$_V$) follows the trigonometric relationship:

$$\text{TDM}_H : \text{TDM}_V = \mu_0^2 \sin^2\varphi_L : \mu_0^2 \cos^2\varphi_L, \quad (1)$$

where $\mu_0$ is the magnitude of the dipole moment and squares of the components indicate the intensity of the transition (emission intensity). The EDO describes an average fraction of the horizontal and vertical dipole moment of whole emitters embedded in the emissive layer. An ensemble average of

the horizontal dipole moment gives the fraction of horizontal emitting dipole moment in the emissive layer ($\Theta$) as a parameter of the EDO by

$$\Theta = \langle \sin^2 \varphi_L \rangle. \qquad (2)$$

Details about the rotation matrix and the vector transformation are given in Methods section.

The deposition simulation was performed by dropping a target molecule onto organic substrates under vacuum followed by thermal equilibration at 300 K as shown in Fig. 1c. The simulations were performed using the Materials Science Suite (Version 2.2) released by Schrödinger Inc.[25] Force field of OPLS_2005[26] and periodic boundary conditions were used for the MD simulations. Organic substrates were prepared by packing of 256 host molecules at 300 K and 1 atm. The simulated substrates have random molecular orientations and similar densities as the experimental results. Detailed steps of preparation of the substrates are given in Supplementary Fig. 1 and Supplementary Note 1. One of the challenges of a single-trajectory-based MD analysis for orientation during deposition is that the time scale needed to observe the entirety of lateral degrees of freedom for a single molecule is much longer than that of a typical MD simulation. As such, we introduced 50 independent deposition events per dopant, instead of relying upon a single MD trajectory for each. About 50 target dopant molecules were distributed in the vacuum slab of the periodic substrate model at un-overlapped locations with different orientations for the deposition simulation. Each target molecule was individually dropped onto the substrate under vacuum at 300 K. Translational motion of the host molecules at the bottom of the substrate was restrained in order to avoid the drift of the system. The deposition simulation used an NVT ensemble for a duration of 6000 ps with a time step of 2 fs and configurations of the system were recorded every 6 ps. One example of the process is shown in Supplementary Movies 1 and 2. Finally, EDOs of the phosphors and the molecular angles ($\varphi_C$) were analyzed using Eq. (1) from the configurations. The analysis is based upon an assumption that the characteristic time to determine the orientation of dopants is in the same scale of which the intermolecular interaction converges after the deposition of a dopant.

**Materials**. Chemical structures of the materials used in this study are depicted in Fig. 2a, b. Three heteroleptic iridium complexes of Ir(ppy)$_2$tmd, Ir(3′,5′,4-mppy)$_2$tmd[18], and Ir(dmppy-ph)$_2$tmd[19] possessing high $\Theta$ values were adopted to investigate the effect of the phosphor molecular structure. The molecular $C_2$ symmetry axis toward the center of the ancillary ligand from the origin located at the Ir atom was set as $\boldsymbol{m}_z$, the orthogonal vector to $\boldsymbol{m}_z$ normal to the molecular Ir-O-O plane was set as $\boldsymbol{m}_x$, and $\boldsymbol{m}_y$ was determined by a cross product of $\boldsymbol{m}_z$ and $\boldsymbol{m}_x$ in the dopants. The triplet TDM vectors of the three Ir complexes align along the direction of the iridium atom to the pyridine rings by $^3$MLCT as displayed in Fig. 2a. Coordinates of the TDM vectors of Ir(ppy)$_2$tmd, Ir(3′,5′,4-mppy)$_2$tmd, and Ir(dmppy-ph)$_2$tmd were $[\varphi_M = 88°, \theta = 147°]$, $[\varphi_M = 89°, \theta = 141°]$, and $[\varphi_M = 89°, \theta = 156°]$, respectively, indicating that the substituents at the 4-position of the pyridine of the main ligands do not change the direction of triplet TDM vectors much.

Diphenyl-4-triphenylsilyphenyl-phosphineoxide (TSPO1), 1,4-bis(triphenylsilyl)benzene (UGH-2), and 4,4′-bis(N-carbazo-lyl)-1,1′-biphenyl (CBP) were selected as host materials to investigate the effect of ground state dipole and conjugation length of the host on the EDO. TSPO1 has large permanent dipole moment due to the polar phosphine oxide group and the asymmetric structure, while UGH-2 and CBP molecules have small ground state dipole moments compared to TSPO1 due to the symmetric structures and the less polar groups. On the other hand, CBP has longer conjugation length than UGH-2 and TSPO1, indicating that CBP has a larger polarizability than UGH-2.

Experimentally, Ir(ppy)$_2$tmd exhibited the $\Theta$ values of 0.60, 0.75, and 0.78 when doped in the UGH-2, CBP, and TSPO1 layers, respectively. Ir(3′,5′,4-mppy)$_2$tmd and Ir(dmppy-ph)$_2$tmd doped in TSPO1 layers have enhanced horizontal dipole orientation with the $\Theta$ values of 0.80 and 0.86, respectively (Supplementary Fig. 2, Supplementary Note 2, and ref. [23]).

**Simulation results**. The simulated variations of the orientation of the TDM$_H$ and $C_2$ axis of the dopants with time on the different hosts are displayed in Supplementary Fig. 3 for 50 depositions for each system. The orientation of the phosphors was stabilized after certain time for some molecules, but fluctuated continuously for other molecules.

Figure 3a exhibits the histograms of the EDO resulting from the deposition simulation. The *blue lines* represent the probability density of TDM$_H$ ($\sin^2 \varphi_L$, derivation in Methods section) of an arbitrary vector. The *green lines* exhibit the deviations of the population from the random distribution. The simulated $\Theta$ values of Ir(ppy)$_2$tmd were 0.63, 0.72, and 0.74 on the UGH-2, CBP, and TSPO1 substrates, respectively. Ir(3′,5′,4-mppy)$_2$tmd and Ir(dmppy-ph)$_2$tmd on TSPO1 substrates have the $\Theta$ values of 0.76 and 0.82, respectively. In addition, the simulation was performed for Ir(ppy)$_3$, a homoleptic complex exhibiting isotropic EDO when doped in CBP as a refs [14, 23]. The distribution of the emitting dipole moment of Ir(ppy)$_3$ was close to the random distribution with a simulated $\Theta$ value of 0.67 and random orientation of the $C_3$ symmetry axis of the molecule (Supplementary Fig. 4; Supplementary Note 3). The simulated EDOs match well with the experimental results as compared in Table 1, verifying that the MD simulation describes the vacuum deposition adequately. The results show that Ir(ppy)$_2$tmd in UGH-2 has larger molecular population with vertical TDM at the expense of reduced population with horizontal TDM compared to the random distribution ($\Theta = 0.67$). Higher $\Theta$ values are obtained when population of molecules possessing high TDM$_H$ is getting larger with the reduced population with low TDM$_H$.

The orientation of the $C_2$ axes of the phosphors on the organic layers is shown in Fig. 3b to find out if alignments of the aliphatic ancillary ligands have any correlation with EDO, for instance, if the horizontal EDO results from the vertical alignment of ancillary ligands with respect to the substrate[20, 24]. One expects that the distribution function follows $\sin \varphi_c$ (*blue line*) if the orientation is random. We can extract several interesting results from Fig. 3b. First, the orientation of the ancillary ligand of the Ir complexes has broad distributions for all the deposited films. Second, host effect on EDO is independent of alignments of the ancillary ligand. The total distribution of the $C_2$ axis of Ir(ppy)$_2$tmd is similar on the UGH-2, CBP, and TSPO1 layers with the average $\varphi_C$ of 70°, but the EDO on UGH-2 host is different from the EDOs on other two hosts. Third, the orientation of the $C_2$ axes of Ir(3,5′,4-mppy)$_2$tmd and Ir(dmppy-ph)$_2$tmd on TSPO1 are more random (closer to $\sin \varphi_c$) even though they possess higher $\Theta$ values than Ir(ppy)$_2$tmd. The random distributions are observed even in the region with high horizontal alignment of the emitting dipole moment (*green regions* in the stacked histogram with $0.95 \leq$ TDM$_H \leq 1$). Fourth, the dopant molecules with vertical TDM (*red regions* in the stacked

histogram with $0 \leq TDM_H \leq 0.3$) have $\varphi_C$ close to 90° for all the system, indicating that the ancillary ligands align parallel to the surface. All the results show that there is little correlation between the orientation of TDMs and alignment of the ancillary ligands.

## Discussion

The size of the substrates turns out to be large enough to simulate the vacuum deposition of the phosphorescent dyes adequately, as confirmed by the similar results obtained on a larger substrate consisting of 1024 molecules (Supplementary Fig. 5;

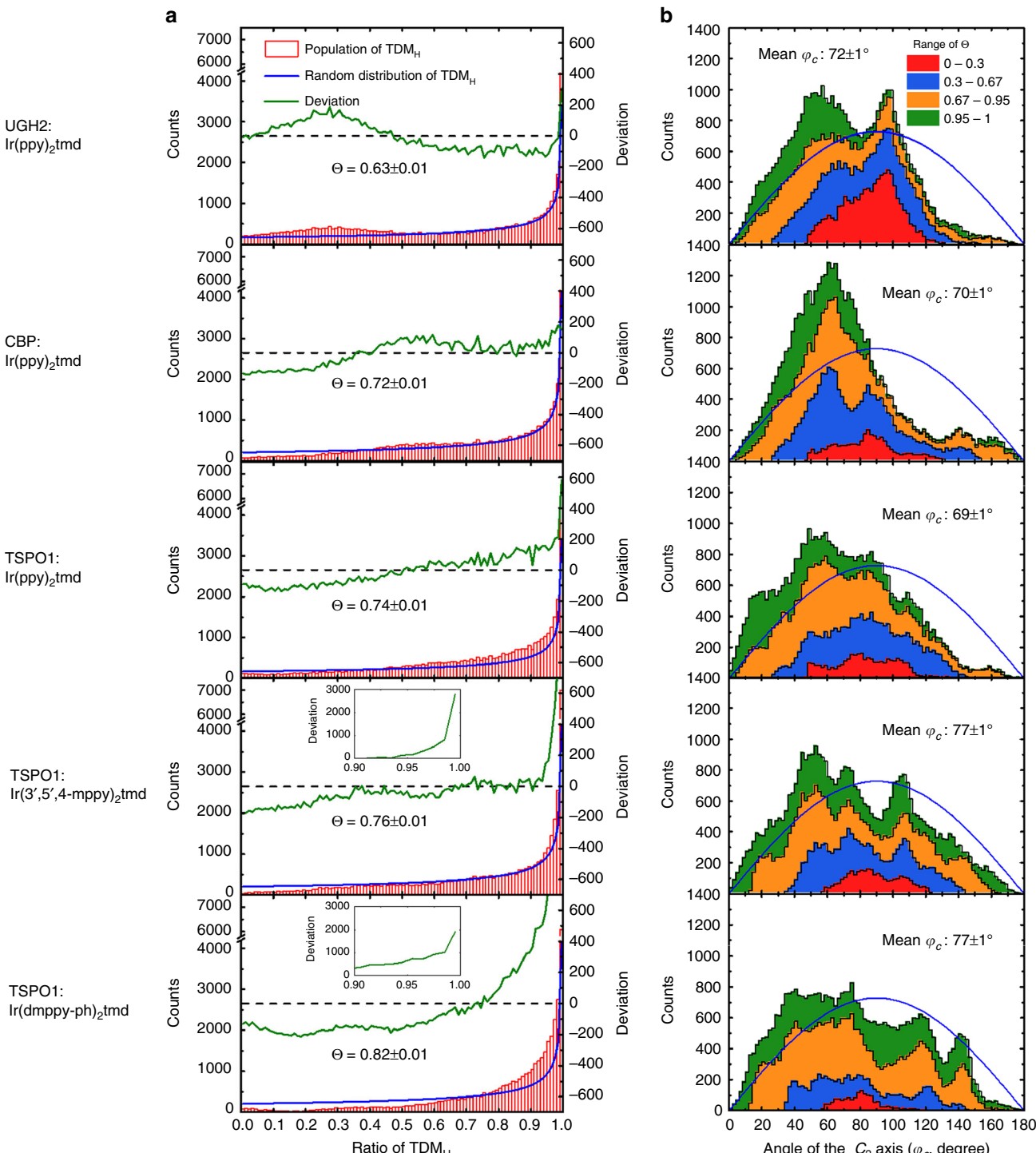

**Fig. 3** Histograms in figure host–dopant combinations from deposition simulation. Each histogram includes 41,700 data in total from the configurations during 50 cases of the deposition in steps of 6 ps in the time regions of 1–6 ns. Data in the time region <1 ns were not used in the statistical analysis to exclude the steps of adsorption and the initial equilibration. **a** Histograms of the EDO with simulated $\Theta$ values. *Red bars* indicate the population of the phosphor configurations having $TDM_H$ values in steps of 0.01. *Blue lines* are theoretical lines of $TDM_H$ from an arbitrary vector of which detailed derivation is given in Methods section. *Green lines* represent deviations of the population compared to the distribution of $TDM_H$ of an arbitrary vector. (*Inset:* enlarged deviation in the region of $0.8 \leq TDM_H \leq 1$) **b** Stacked histogram of the angle of the $C_2$ axis of phosphors and mean angles. Populations of the vector are plotted in steps of 2°. Distribution of the angle in different ranges of $TDM_H$ is distinguished by different *colors*

**Table 1 Comparison of simulated and measured EDOs**

| Host | CBP | UGH-2 | CBP | TSPO1 | TSPO1 | TSPO1 |
|---|---|---|---|---|---|---|
| Dopant | Ir(ppy)$_3$ | Ir(ppy)$_2$tmd | Ir(ppy)$_2$tmd | Ir(ppy)$_2$tmd | Ir(3′,5′,4-mppy)$_2$tmd | Ir(dmppy-ph)$_2$tmd |
| Simulation | 67:33 | 63:37 | 72:28 | 73:27 | 76:24 | 82:18 |
| Measurement | 67:33 | 60:40 | 75:25 | 78:22 | 80:20 | 86:14 |

The result of Ir(ppy)$_3$ doped in the CBP layer is added to the results in five combinations of host and heteroleptic Ir complexes for reference

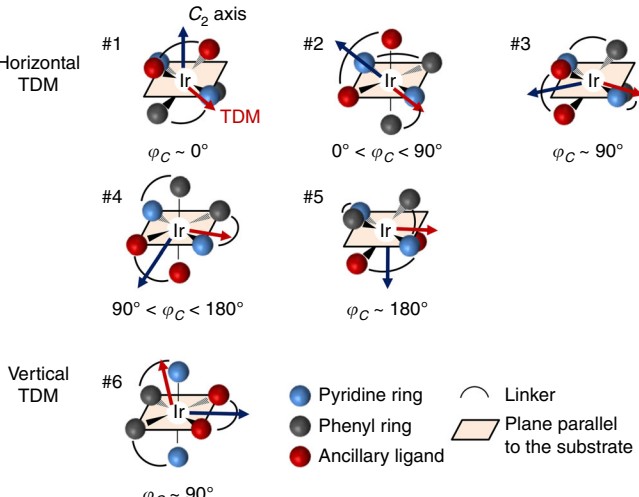

**Fig. 4** Molecular configurations of heteroleptic Ir complexes. The molecular configurations of heteroleptic Ir complexes having horizontal and vertical transition dipole moments are schematically illustrated with *blue*, *gray*, and *red spheres* at the octahedral sites representing pyridine rings, phenyl rings, and the ancillary ligand (–tmd), respectively. *Dark blue* and *red arrows* indicate the molecular $C_2$ axis and the TDM vector, respectively. Five configurations of the molecule for horizontal TDM and one configuration for vertical TDM are illustrated depending on the angle of the $C_2$ axis

Supplementary Note 4). The computation of the autocorrelation times of the molecular angles for the 50 independent depositions for each phosphorescent molecule verifies that the number of events sampled during the deposition (the simulation time and the number of the deposition events) is large enough to validate the MD simulation of the vacuum deposition process (Supplementary Fig. 6; Supplementary Note 5).

The aggregation effect of the Ir complexes is neglected in the MD simulation by depositing each target molecule on an organic substrate at a time and by repeating for 50 molecules deposited on different positions of the organic substrates. The very good consistency of the simulated EDO and experimental values indicates that aggregation is not a necessary condition for the alignment of heteroleptic Ir complexes.

Alignment of aliphatic ligands of heteroleptic Ir complexes to vacuum (model 3) is not required for preferred horizontal EDO either as shown in Fig. 3. A much larger portion of the aliphatic ligand (–tmd group) of Ir(ppy)$_2$tmd molecules align to the vacuum side ($0° < \varphi_C < 90°$ in Fig. 3b) than Ir(3′,5′,4-mppy2)tmd and Ir(dmppy-ph)$_2$tmd molecules. However, the $\Theta$ value of Ir(ppy)$_2$tmd is much lower than Ir(3′,5′,4-mppy)$_2$tmd and Ir(dmppy-ph)$_2$tmd. These results are the reverse direction from the prediction based on the model and clearly demonstrate, therefore, that alignment of aliphatic ligands to the vacuum side is not a necessary condition for the alignment of EDO in heteroleptic Ir complexes. The reason why it is not required can be understood from the following consideration.

The relationship between the orientation of molecules and emitting dipole moment can be easily figured out using schematic molecular orientations of a heteroleptic Ir complex shown in Fig. 4. The $C_2$ axis is toward the ancillary ligand (*dark blue arrows*) and the TDM vector (*red arrows*) is approximately along the direction from the iridium center to one of the pyridine rings. The alignment of iridium-pyridines determines the orientations of TDM for Ir(ppy)$_2$tmd, Ir(3′,5′,4-mppy)$_2$tmd, and Ir(dmppy-ph)$_2$tmd. Figure 4 shows five configurations with different rotation angles of the $C_2$ axis for the horizontal TDM and one configuration for the vertical TDM. Rotation of the $C_2$ axis from the vertical to the horizontal direction can result in the horizontal TDM as long as the TDM is located on the horizontal plane (substrate) with an arbitrary orientation of the ancillary ligand. In other words, horizontal EDO is possible no matter which direction of the ancillary ligand aligns, either toward vacuum or film. On the other hand, the vertical TDM is obtained only when the pyridine rings are aligned perpendicular to the substrate. It accompanies horizontal alignment of the $C_2$ axis ($\varphi_C \sim 90°$) on the configuration. This consideration is consistent with the simulation results in Fig. 3.

The distributions of the ancillary ligand can be partly explained by analyzing the differences in Hildebrand solubility parameters ($\delta$) of the phosphor and host molecules. Predicted solubility parameters of the molecules are 16.2 (UGH-2), 18.5 (CBP), 17.2 (TSPO1), 15.3 (Ir(ppy)$_3$), 14.7 (Ir(ppy)$_2$tmd), 13.8 (Ir(3′,5′,4-mppy)$_2$tmd), and 14.4 MPa$^{1/2}$ (Ir(dmppy-ph)$_2$tmd), respectively, calculated from OPLS_2005 NPT MD by the equation:[27]

$$\delta = \left( \frac{\Delta E_v}{V_m} \right)^{1/2}, \qquad (3)$$

where $\Delta E_v$ is the internal energy change of vaporization, and $V_m$ is the molar volume, respectively. In general, the differences in the solubility parameters ($\Delta \delta$) between two components in a chemical mixture can be an indicator of the degree of miscibility, with smaller and larger values of $\Delta \delta$ indicating more and less miscible, respectively. In this work, the host and the phosphor $\Delta \delta$ are much less than 7 MPa$^{1/2}$, suggesting that all the phosphors are miscible with the hosts[28]. However, $\Delta \delta$'s between Ir(ppy)$_3$ and the hosts are smaller than between Ir(ppy)$_2$tmd and the hosts. Since the difference comes from the ppy and tmd groups, it follows that the tmd group is less miscible in the host substrates than the main ligand ppy group, which explains the orientation of the aliphatic ancillary ligand toward vacuum side for Ir(ppy)$_2$tmd. On the other hand, the difference in the solubility parameters among Ir(ppy)$_2$tmd, Ir(3′,5′,4-mppy)$_2$tmd, and Ir(dmppy-ph)$_2$tmd comes from the difference in main ligands. The reduced solubility of Ir(3′,5′,4-mppy)$_2$tmd and Ir(dmppy-ph)$_2$tmd indicates that both 3′,5′,4-mppy and dmppy-ph groups are less miscible to the host than ppy of Ir(ppy)$_2$tmd and less preference to attachment to the substrate compared to the ppy group. Therefore, the orientations of the ancillary ligand of the two phosphors are more randomized during the deposition, consistent with the simulated distributions in Fig. 3b.

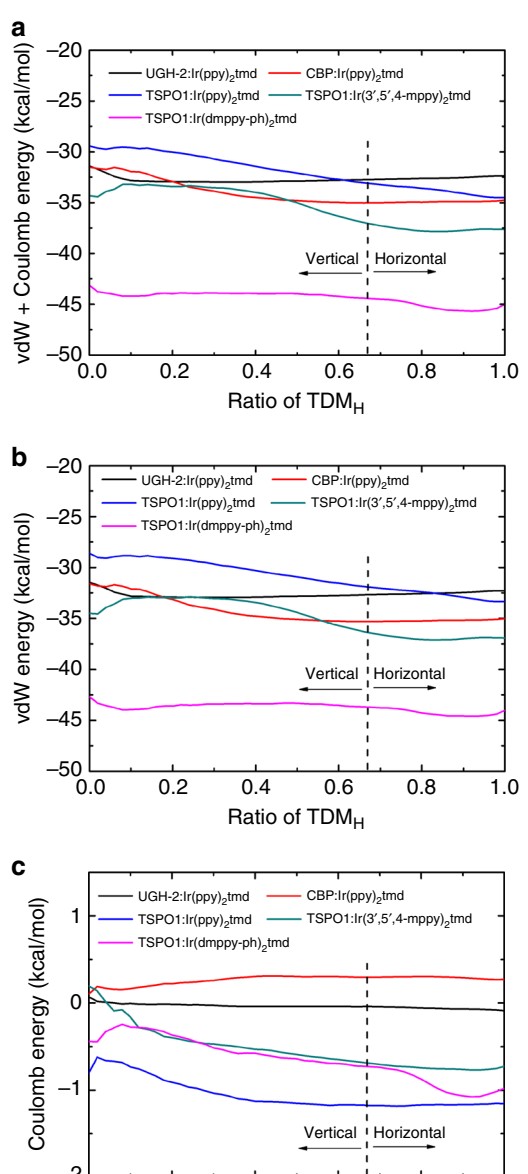

**Fig. 5** Non-bonded interaction energies. **a** Calculated non-bonded van der Waals and Coulomb interaction energies with the cut-off radius of 0.9 nm of each atom of the phosphors as a function of $TDM_H$ in the five host–dopant systems. **b** Van der Waals and **c** Coulomb interaction energies are separated

Non-bonded interaction energy is calculated as a summation of van der Waals and Coulomb interaction energies from the MD simulation to investigate if the intermolecular interaction between the phosphor and neighbor host molecules is responsible for the spontaneous molecular alignments of the phosphors on the surfaces. Figure 5a depicts the correlation between non-bonded interaction energy and orientation of the emitting dipole moment of the phosphors in the five different host-dopant systems. Distributions of the non-bonded interaction energy are given in Supplementary Fig. 7. There is a broad energy trap of ~3 kcal/mol in the region of $TDM_H = 0.1$–$0.5$ for $Ir(ppy)_2tmd$ on the UGH-2 host and the energy increases with further increasing of $TDM_H$, thereby resulting in rather a vertical EDO compared to random orientation because the population of $TDM_H$ is expected to be concentrated in the regions of low (large) non-bonded interaction

energy. On the other hand, non-bonded interaction energies of $Ir(ppy)_2tmd$ on CBP and TSPO1 layers, and the energies of $Ir(3',5',4\text{-mppy})_2tmd$ and $Ir(dmppy\text{-ph})_2tmd$ on TSPO1 are lowered as $TDM_H$ increases. As a result, molecular alignment with horizontal TDM is energetically preferred when they are deposited onto the organic semiconducting layers. Furthermore, much lower energies were obtained from $Ir(3',5',4\text{-mppy})_2tmd$ and $Ir(dmppy\text{-ph})_2tmd$ than $Ir(ppy)_2tmd$ on the TSPO1 layer, indicating that the increased EDOs are also related to the stabilization by neighbor molecules. The calculated non-bonded interaction energy and the statistical results indicate that the host–dopant interaction plays a pivotal role in orienting hetero-leptic Ir complexes and the force applies to the alignment of the iridium–pyridine bonds of the phosphors toward the horizontal direction.

The type and magnitude of the non-bonded interactions are different for different phosphors and hosts, leading to different EDOs. The separated vdW and Coulomb energies as a function of the molecular orientation are depicted in Fig. 5b, c. The vdW energies in $CBP:Ir(ppy)_2tmd$, $TSPO1:Ir(ppy)_2tmd$, $TSPO1:Ir(3',5',4\text{-mppy})_2tmd$, and $TSPO1:Ir(dmppy\text{-ph})_2tmd$ systems decrease, whereas the energy in $UGH\text{-}2:Ir(ppy)_2tmd$ increases as the ratio of the horizontal transition dipole moment ($TDM_H$) increases. For a polar host molecule of TSPO1, Coulomb energies of the phosphors are lowered as $TDM_H$ increases. The variation of vdW energy depending on the molecular orientation was 3–5 kcal/mol, which is larger than the variation of Coulomb energy of 0–1 kcal/mol. The results indicate that vdW interactions (dipole-induced dipole and induced dipole-induced dipole interactions) between the aromatic ligands and the nearest host molecules are the main mechanism contributing to the molecular alignment of the phosphors. The Coulomb interaction helps further alignments of the phosphors if polar host materials are employed. For instance, $Ir(ppy)_2tmd$ and $Ir(3',5',4\text{-mppy})_2tmd$ have a quadrupole composed of two dipoles from pyridines ($\delta+$ charge) to Ir atom ($2\delta-$ charge). If there is a dipole in host molecule (i.e., TSPO1), dipole and quadrupole interaction [$-P=O^{\delta-}$ and $^{\delta+}H(pyridine)$] anchors one phosphor molecule to two host molecules, leading to a rather horizontal orientation of iridium–pyridines bond of the phosphors, which is approximately parallel to the TDM. In contrast, if host molecule has the positive surface potential (i.e., $^{\delta+}(phenyl)_3\text{-}Si\text{-}phenyl\text{-}Si\text{-}(Phenyl)_3^{\delta+}$ in UGH-2), there must be a repulsive force between pyridine of phosphors and host molecules so that pyridine ring must be pushed to vacuum. Dispersion force between the conjugated phenyl substituents of $Ir(dmppy\text{-ph})_2tmd$ and nearest neighbors anchors the pyridines onto the surface as well and lowers the energy with the molecular long axis lying on the surface. Meanwhile, random EDO of $Ir(ppy)_3$ is attributed to three intermolecular interaction sites, resulting in random orientation of the molecule.

Figure 6a–c exhibit the representative molecular behaviors of phosphors and nearest host molecules during the deposition out of 50 cases (Supplementary Fig. 3) with different host–dopant combinations of $UGH\text{-}2:Ir(ppy)_2tmd$, $TSPO1:Ir(ppy)_2tmd$, and $TSPO1:Ir(dmppy\text{-ph})_2tmd$, respectively. Large vibrations, rotations, and diffusions of $Ir(ppy)_2tmd$ on the surface of the UGH-2 layer without lowering the energy were observed in the trajectory shown in Fig. 6a. The perpendicular alignment of pyridines occasionally formed on the surface resulted in vertical emitting dipole moment in average. On the other hand, a hydrogen atom at one of pyridines of $Ir(ppy)_2tmd$ faced toward an oxygen atom of TSPO1 with the $-P=O\cdot\cdot H(pyridine)$ distance around 0.4 nm at $t = 2472$ and $4560$ ps, thereby the parallel alignment of the Ir-pyridines of $Ir(ppy)_2tmd$ to the surface. Larger quadrupole moment of $Ir(3',5',4\text{-mppy})_2tmd$ than that of

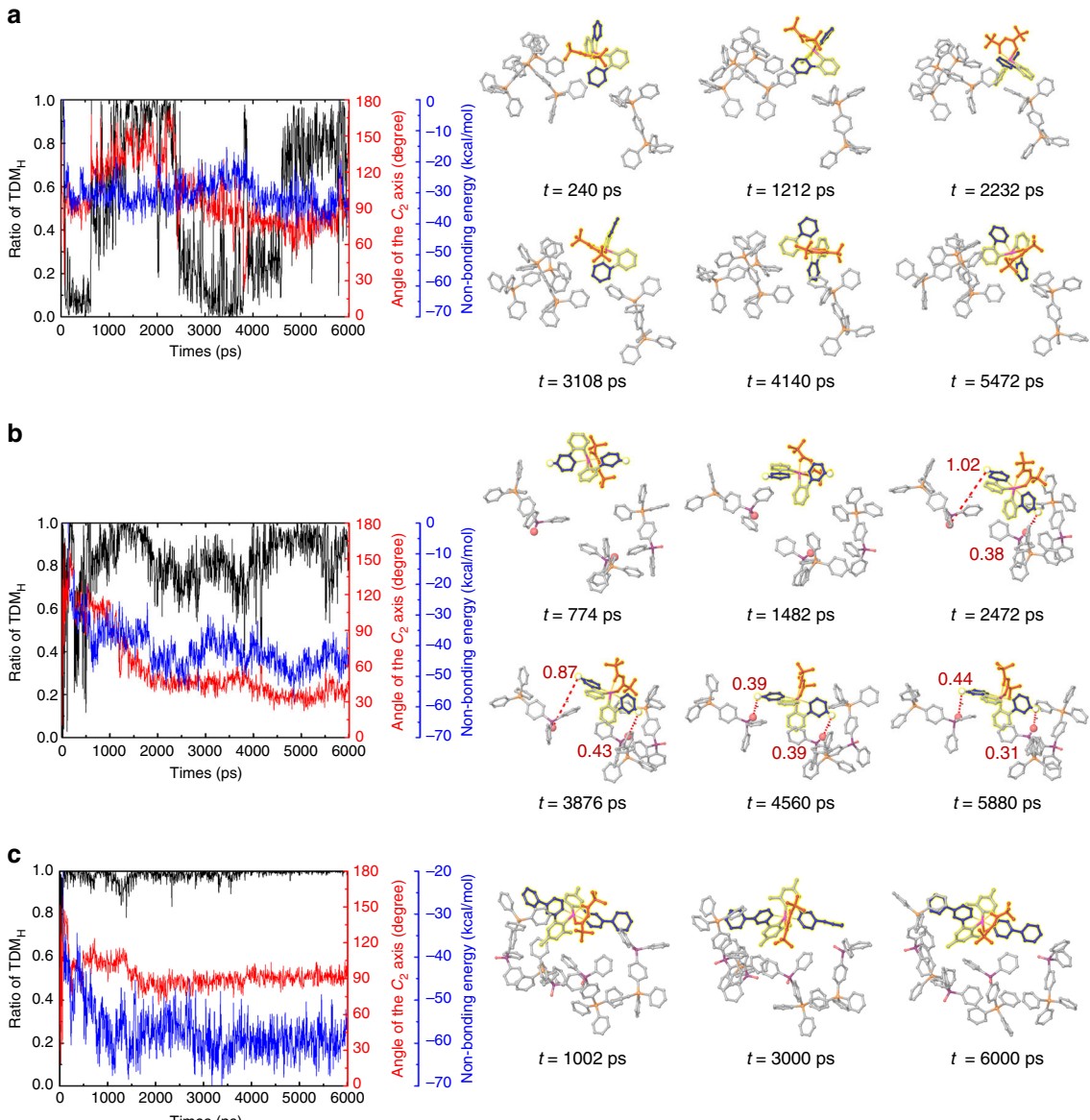

**Fig. 6** Representative configurations during deposition. Snapshots of local configurations and time-dependent trajectories of the EDO, angle of the $C_2$ axis, and non-bonded interaction energy up to 6 ns are depicted together. The ancillary ligand and pyridine rings of the phosphors at the octahedral sites are colored by *red* and *blue*, respectively. **a** Ir(ppy)$_2$tmd deposited onto the UGH-2 layer has a continuous rotation and the occasionally observed perpendicular alignment of pyridines with respect to the substrate results in vertical EDO. **b** Ir(ppy)$_2$tmd anchors on the surface of TSPO1 layer by the local quadrupole–dipole interaction between the two nearest host molecules located at both sides. The hydrogen atoms at both pyridines of Ir(ppy)$_2$tmd and the oxygen atoms of TSPO1 connected by a *broken line* were the plausible binding sites. The distances between the two atoms (*broken lines*) are getting closer until around 0.4 nm as the time increases and a host-dopant–host pseudo complex is formed with the parallel alignment of pyridines with respect to the substrate. **c** Ir(dmppy-ph)$_2$tmd deposited onto the TSPO1 layer are less mobile than Ir(ppy)$_2$tmd with the low non-bonded interaction by the configuration of large dispersion force energy along the direction of TDM

Ir(ppy)$_2$tmd increased the strength of quadrupole–dipole interaction and resulted in enhancement of fraction of the horizontal dipole compared to Ir(ppy)$_2$tmd. The horizontal EDO of Ir(ppy)$_2$tmd in CBP could be understood by the dipole inducement in carbazole groups of CBP when the positive pole of pyridine approaches, but the interaction strength for the iridium–pyridines alignment between Ir(ppy)$_2$tmd-CBP is smaller than that between Ir(ppy)$_2$tmd–TSPO1. Compared to the former cases, the picture of Ir(dmppy-ph)$_2$tmd shown in Fig. 6c is rather simple. The phosphor deposited onto the TSPO1 layer was stabilized after short time of deposition to the one with the horizontal iridium-pyridine-phenyl alignment and maintain the configuration. The much lower (larger) non-bonded interaction energy of

Ir(dmppy-ph)$_2$tmd in Fig. 5a restrained rotation of the molecule and the molecular configurations are easily fixed on the surface, resulting in much enhancement of horizontal EDO was achieved by the substitutions.

The origin of the molecular orientation and EDO of doped heteroleptic iridium complexes in vacuum-deposited organic layers is investigated using MD simulations and quantum mechanical analyses in direct comparison with experimental observation. Careful analyses of the simulation results revealed that molecular alignments of the phosphors are spontaneous by local electrostatic and van der Waals interaction with nearest host molecules interacting in a smaller scale than a molecule. The orientation of the TDM vector of the phosphors on the organic

surfaces follows the direction of the ligand mainly participating in optical transition, such as pyridines in ppy, in the molecular alignment, whereas the alignment of ancillary ligand does not have a direct correlation with the EDO. Attractive interactions between pyridines of a phosphor and CBP (quadrupole-induced dipole interaction) or TSPO1 (quadrupole–dipole interaction) anchor the phosphor onto host molecules with the parallel iridium-pyridine alignment, thereby increasing the horizontal EDO. Ir(3′,5′,4-mppy)$_2$tmd has larger quadrupole moment than Ir(ppy)$_2$tmd resulted in further molecular alignment for the horizontal emitting dipole moment. The increase of the dispersion force along the direction of TDM was also effective on control of the molecular orientation for the horizontal EDO with lowered non-bonded interaction energy.

## Methods

**Quantum mechanical calculations and molecular dynamics simulations**. Density functional theory (DFT) was used to obtain molecular geometries and electrostatic potentials of host and phosphors. Triplet TDMs of the same phosphors from $T_1$ to $S_0$ were also calculated via SOC-TDDFT. The DFT and SOC-TDDFT calculations and the follow-up analyses were performed with Schrodinger Materials Science Suite[25] along with the quantum chemical engine, Jaguar[29]. A TDM having the largest oscillator strength among the three degenerated states of $T_1$ level ($T_x$, $T_y$, and $T_z$) obtained from the density functional calculations was used in this study as a representative TDM. All MD simulations were performed by Desmond[30, 31], a MD engine implemented in the Schrodinger Materials Science Suite. Equilibration simulations prior to deposition were performed in NPT ensembles, where pressure and temperature were set constant via Nose–Hoover chain and Martyna–Tobias–Klein method, respectively. There were no explicit constraints to geometry and/or positions to any of the molecules that were introduced in the simulation box. The simulations were performed over NVIDIA general-purpose GPU cards (K80).

**Rotation matrix method**. Rotation matrix method was used for transformation of the TDM vector from the molecular coordinate to the laboratory coordinate. Rotation angles of $\alpha$, $\beta$, and $\gamma$ are defined as the clockwise rotations to laboratories axes of $n_x$-, $n_y$-, and $n_z$-axes, respectively. Then, the rotation matrixes for $\alpha$, $\beta$, and $\gamma$ rotations are followings:

$$R_\alpha = \begin{pmatrix} 1 & 0 & 0 \\ 0 & \cos\alpha & \sin\alpha \\ 0 & -\sin\alpha & \cos\alpha \end{pmatrix}, \tag{4}$$

$$R_\beta = \begin{pmatrix} \cos\beta & 0 & -\sin\beta \\ 0 & 1 & 0 \\ \sin\beta & 0 & \cos\beta \end{pmatrix}, \tag{5}$$

$$R_\gamma = \begin{pmatrix} \cos\gamma & \sin\gamma & 0 \\ -\sin\gamma & \cos\gamma & 0 \\ 0 & 0 & 1 \end{pmatrix}. \tag{6}$$

Sequential $\alpha\beta\gamma$ rotations of the dopant molecules were extracted in every configurations of the MD simulation. A product of the three rotation matrixes gives a matrix representing the orientation of the dopant molecule by

$$R_{total} = R_\gamma R_\beta R_\alpha. \tag{7}$$

Finally, the TDM vectors in the laboratory coordinate were obtained by

$$TDM_{Lab} = R_{total} TDM_{Mol}. \tag{8}$$

**Calculation of a probability density function of TDM$_H$**. Integration of a probability density function, $f$, indicates a probability of a variable $X$ between

$X = a$ and $b$.

$$P(a<X<b) = \int_a^b f_X(x)dx, \tag{9}$$

where

$$\int_{-\infty}^{\infty} f_X(x)dx = 1. \tag{10}$$

To calculate the probability density function of $\sin^2\varphi$ from an arbitrary vector, we define an arcsine function of

$$y = \arcsin(\sqrt{x}), \ 0 \leq x \leq 1, \tag{11}$$

which is a reversed function of $y = \sin^2 x$. For a monotonic function, the variables are related by

$$f_X(x) = f_Y(y)\frac{dy}{dx}. \tag{12}$$

If we put $f_Y(y) = \sin(y)$ and $\frac{dy}{dx} = \frac{1}{2\sqrt{x-x^2}}$ into equation (A4), the probability density function ($f_X$) is obtained as

$$f_X(x) = \frac{1}{2\sqrt{1-x}}. \tag{13}$$

**Data availability**. The authors declare that all data supporting the findings of this study are available in the article and in Supplementary Information file. Additional information is available from the corresponding author upon request.

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

## Acknowledgements

This work was supported by the Midcareer Research Program through an NRF (National Research Foundation) grant funded by the MSIP (Ministry of Science, ICT, and Future Planning) (2014R1A2A1A01002030). We appreciate the helpful comments from Drs Mathew D. Halls and H. Shaun Kwak from Schrödinger Inc., Prof. Youn Joon Jung from Seoul National University, and Dr. Denis Andrienko from Max Planck Institute for Polymer Research (MPIP).

## Author contributions

C.-K.M. prepared samples, designed experiments, demonstrated calculations and simulations, prepared mathematical models, and prepared the manuscript; K.-H.K. prepared materials; J.-J.K. directed experiments, calculation, and the manuscript.

## Additional information

**Competing interests:** The authors declare no competing financial interests.

