## [Peer Review File · Nature Communications]

Reviewers' comments:

Reviewer #1 (Remarks to the Author):

In this manuscript, the authors perform the MD simulation to derive the origin of the molecular orientation of phosphors in the emissive doped films in OLEDs. They carefully investigate the relationship in the directions of C2 axis (ancillary ligand), main ligand (cyclometalating ligand), emission transition dipole moment, and surface normal by step-by-step sequential simulations, and conclude the origin of the phosphor orientation. Though some of their conclusions have already been discussed in some past paper, some other conclusions are new and very informative for understanding the detail of the mechanism. In addition, this research clearly visualizes what occurs at the molecular level with the quantification of energy and time, and also directly support what have been discussed in past. I feel that this research is very interesting and meets the level of the publication in Nature Communication after revision.

The points that should be revised are as follows. The main ones are (1)-(3).

(1) Methods

The authors say that the MD simulations are performed using software, but many parts in the simulations are not sufficiently explained and are in a black box because the software is not very popular. The authors should describe the fundamental information and the main assumptions in the simulations: for example, how are the molecular structures calculated (including the calculation level), how is the molecular structures in their MD simulation (fixed or flexibly changed), how is the intermolecular interaction estimated, and etc. The description of the full information may make the manuscript redundant, but the "fundamental" information and the "main" assumptions should be shown properly.

(2) Ir(ppy)₃

The authors discuss the difference between the molecules and Ir(ppy)₃ in some parts in the manuscript. I think that they should simulate the case of Ir(ppy)₃ and directly compare its result to others, if they would like to conclude the origin of the difference. This simulation also guarantee the validity of their simulations. Because the dipole orientation of Ir(ppy)₃ is almost random ($\theta \sim 0.67$), it will be a nice reference.

(3) Conclusion

In the abstract, the authors say "Parallel alignment of the main cyclometalating ligands in the molecular complex due to host interactions rather than the ancillary ligand orienting to vacuum leads to the horizontal EDO". However, in Line 199, they also say "tmd group is less miscible in the host substrates than the main ligand ppy group, which explains the orientation of the aliphatic ancillary ligand toward vacuum side for Ir(ppy)₂tmd". The latter means that the small miscibility of the ancillary ligand also contributes to the horizontal orientation of the emission dipole moments. The authors should consider what is the principal effect and what is the secondary effect, and carefully reconstruct the logic.

(4) Line 113

The definitions of "mx" and "my" seem opposite. In Fig. 2a, mx is normal to the molecular Ir-O-Ir plane, not in the the molecular Ir-O-Ir plane.

(5) Line 116

"Fig. 2b" should be "Fig. 2a".

(6) Line 136 and Fig. S3

Fig. S3 should have a proper caption (explanation) in the Supporting Information file. The meanings of the black and red curve are not explained here. (They can be understood in Fig. 5 later.)

(7) Line 139

I would like to ask the authors to give a more detailed explanation how the blue curve and red bars in Fig. 3a are calculated.

(8) Line 178

"6 configurations" should be "5 configurations"?

(9) Line 213 and 221

It is not easy to read the energy differences from Fig. S4. I recommend that the author revise Fig. S4 and/or show the values in the main text.

(10) Line 250

"Fig. 4a" should be "Fig. S4"?

Reviewer #2 (Remarks to the Author):

The authors attempt to explain molecular orientations of Ir complexes in an organic host using molecular dynamics simulations. The topic is definitely interesting and timely, however the paper is rather technical and to some extent inconclusive (stating that non-bonded interactions guide the orientation does not really help). I suggest publishing in a more specialized journal.

In addition, I have reservation with respect to the methods used in the manuscript (see below) which the author might take into account when resubmitting.

1. The description of the MD protocol is incomplete. Was the film annealed between two deposition events? Was the rest of the molecules fixed during the annealing protocol? The supplied movie shows that there is no lateral molecular dynamics. To me this is a clear indication that the system is not equilibrated long enough between deposition events.

2. The authors are using a commercial MD package with the built-in OPLS force-field. This force-field was never parametrized for iridium complexes and conjugated compounds. It is therefore difficult for me to judge how trustable the simulations are.

3. Systems size are small. I believe that one needs much better statistics to make quantitative conclusions. The error bars for averaged angles (which are very similar) are not provided.

4. I do not think that the solubility parameters are useful to predict molecular orientations. We are dealing with the organic-vacuum interface: The processes here are very different from the bulk (2D diffusion, interfacial dipole moments, etc). I do not see a physical reason why solubility should correlate to molecular orientations.

Reviewer #3 (Remarks to the Author):

This work mainly describes the orientation of phosphors like Ir-complexes and its influences on an enhancement of outcoupling efficiency of light. What I concerned is the following two points:

1. The crystal structure of the first Ir-complex $[\text{Ir}(\text{ppy})_2(\text{tmd})]$ containing water was reported. Please the authors to address what the orientation in crystalline $[\text{Ir}(\text{ppy})_2(\text{tmd})]$ as phosphors in in-plane and out of plane. Also add their XRD patterns (in- and out-plane) into text to support your claims.
2. In introductions, the molecular orientation of thin film is essential in physical properties especially extremely anisotropic properties: dielectric, ferroelectric. So, the authors should cite some of works relative to this field.
3. I encouraged the authors to determine three Ir-complex crystal structures.
After done, this is very nice paper.

Reviewers' comments:

Reviewer #1 (Remarks to the Author):

I think that the revisions by the authors were made appropriately. However, regarding Comment #3, the authors completely deleted one long paragraph in their discussions, which is essentially related to the orientation mechanism. They only deleted a paragraph and did not add any explanation for understanding the mechanism. I think that they should clearly explain the mechanism of "parallel alignment of the main cyclometalating ligands" in more detail to support their conclusion.

Reviewer #3 (Remarks to the Author):

The authors have addressed the concerns raised in the previous round of review, and I agree their explanation. The work is important and the results are convincing. Therefore, the paper is now appropriate for publication.

Reviewer #4 (Remarks to the Author):

- 1) I agree with reviewer 2 that the number of events sampled is very small and it is unclear whether the sampling of the trajectory actually helps, because the authors state that some molecules remain fixed (i.e. they contribute always the same value across the whole trajectory), while other molecules change conformation. In order to quantify this, it would be nice to compute the autocorrelation time of the angle and plot a histogram of this for the 50 molecules. This should tell us the average number of independent conformations during the trajectory.
- 2) On the positive side, the agreement between the experimental and the theoretical values is amazingly good.
- 3) I think the manuscript in the present form is lacking a message beyond the fact that the protocol seems to work. I concur with the reviewer on point 1 that stating that "nonbonded interactions" are responsible in a system where no new bonds are formed or broken is not really illuminating. In their response, the authors claim that they can distinguish aliphatic from electrostatic mechanisms in their reply to the reviewer and this could be easily demonstrated by comparing the relative contribution of the nonbonded contributions of the vdW energy and electrostatic energy as a function of orientation which their program can just print out. As a function of orientation we should have an energy profile for both contributions to the nonbonded energy and if the authors are correct, the variation of the vdW energy should be weaker than that of the Coulomb energy.
- 4) Going beyond the comments of reviewer 2 in this respect, my worry is that the aliphatic interaction with a substrate of just 256 molecules is too weak. The interaction falls like r^{-6} with distance, but integrated over the bulk, this makes a huge contribution. Therefore the contribution of the vdW interactions may be underestimated by the present model of the substrate. The authors could repeat the 50 simulations for one system with 1024 molecules as a substrate and demonstrate that the variation in the vdW energy does not change significantly

REVIEWERS' COMMENTS:

Reviewer #1 (Remarks to the Author):

I think that the revisions were made appropriately. Although the discussion on the solubility is not weighted, the authors mention it carefully in the Supplementary Information. I would like to recommend the publication of this nice work.

Reviewer #4 (Remarks to the Author):

I think all my questions have been appropriately addressed.

Detailed response to the reviewers' comments

Title: Unraveling the orientation of phosphors doped in organic semiconducting layers

Author: Jang-Joo Kim; Seoul National University

We appreciate the reviewers' valuable comments and we revised the manuscript. Revised words or sentences in response to comments by reviewer #1, #2, and #3 were highlighted by yellow, light blue, and light green colors, respectively, both in the manuscript and Supplementary Information.

Followings are the detailed responses to the reviewers' comments.

Reviewer #1 (Remarks to the Author):

In this manuscript, the authors perform the MD simulation to derive the origin of the molecular orientation of phosphors in the emissive doped films in OLEDs. They carefully investigate the relationship in the directions of C2 axis (ancillary ligand), main ligand (cyclometalating ligand), emission transition dipole moment, and surface normal by step-by-step sequential simulations, and conclude the origin of the phosphor orientation. Though some of their conclusions have already been discussed in some past paper, some other conclusions are new and very informative for understanding the detail of the mechanism. In addition, this research clearly visualizes what occurs at the molecular level with the quantification of energy and time, and also directly support what have been discussed in past. I feel that this research is very interesting and meets the level of the publication in Nature Communication after revision.

The points that should be revised are as follows. The main ones are (1)-(3).

Comment 1. Methods

The authors say that the MD simulations are performed using software, but many parts in the simulations are not sufficiently explained and are in a black box because the software is not very popular. The authors should describe the fundamental information and the main assumptions in the simulations: for example, how are the molecular structures calculated (including the calculation level), how is the molecular structures in their MD simulation (fixed or flexibly changed), how is the intermolecular interaction estimated, and etc. The description of the full information may make the manuscript redundant, but the "fundamental" information and the "main" assumptions should be shown properly.

Reply: We fully acknowledge the need of including all relevant information regarding the MD simulation so that the results reported in this work are reproducible and the analysis is transparent. As such, we added detailed information regarding MD simulations performed in this work (Please note that the most simulation details needed to reproduce the results – simulation time for host equilibration and dopant deposition, as well as time steps and sampling rates – are listed in the last paragraph before Materials section).

Readers will be able to reach the same conclusion using any MD simulation codes given the same conditions were assessed. We also note that Desmond, the molecular dynamics engine used in this work, is a popular MD code among scientists who are working in a wide array of applications ranging from materials science to life science. A major reason that Desmond was used in this work is its efficiency and speed utilizing the latest GPU technology in MD integration and analysis. However, the scientific results reported in this work should be reproducible by any molecular dynamics and analysis tools given enough time and resource to produce the dataset based on the same force field.

The last sentence of the first subsection of the Methods section were replaced by following paragraphs.

Revision: Method section on page 14, line 287.

Before –

MD simulations were carried out on an NVIDIA GPU using the Desmond molecular dynamics engine.²⁸⁻²⁹

After –

All MD simulations were performed by Desmond,²⁸⁻²⁹ a molecular dynamics engine implemented in the Schrodinger Materials Science Suite. Equilibration simulations prior to deposition were performed in NPT ensembles where pressure and temperature were set constant via Nose-Hoover chain and Martyna-Tobias-Klein method, respectively. There were no explicit constraints to geometry and/or positions to any of the molecules that were introduced in the simulation box. The simulations were performed over NVIDIA general-purpose GPU cards (K80).

Comment 2. Ir(ppy)₃

The authors discuss the difference between the molecules and Ir(ppy)₃ in some parts in the manuscript. I think that they should simulate the case of Ir(ppy)₃ and directly compare its result to others, if they would like to conclude the origin of the difference. This simulation also guarantees the validity of their simulations. Because the dipole orientation of Ir(ppy)₃ is almost random (theta~0.67), it will be a nice reference.

Reply: We appreciate the reviewer for the valuable comment. We performed the simulation for Ir(ppy)₃ and its emitting dipole orientation turned out to be random indeed with the theta value of 0.67. We have added the results of Ir(ppy)₃ in main text and Supplementary Information with an additional figure.

Revision: Table 1 on page 29 and caption on page 23.

Before –

Table 1. Comparison of simulated and measured EDOs in 5 combinations of the host and the dopant molecules.

Host	UGH-2	CBP	TSPO1	TSPO1	TSPO1
Dopant	Ir(ppy) ₂ tmd	Ir(ppy) ₂ tmd	Ir(ppy) ₂ tmd	Ir(3',5',4- mppy) ₂ tmd	Ir(dmppy- ph) ₂ tmd
Simulation	63:37	72:28	73:27	76:24	82:18
Measurement	60:40	75:25	78:22	80:20	86:14

After –

Table 1. Comparison of simulated and measured EDOs in 5 combinations of the host and the heteroleptic Ir complexes in addition to Ir(ppy)₃ doped in the CBP layer for reference.

Host	CBP	UGH-2	CBP	TSPO1	TSPO1	TSPO1
Dopant	Ir(ppy) ₃	Ir(ppy) ₂ tmd	Ir(ppy) ₂ tmd	Ir(ppy) ₂ tmd	Ir(3',5',4- mppy) ₂ tmd	Ir(dmppy- ph) ₂ tmd
Simulation	67:33	63:37	72:28	73:27	76:24	82:18
Measurement	67:33	60:40	75:25	78:22	80:20	86:14

Addition: On page 7, line 149 of main text.

In addition, the simulation was performed for Ir(ppy)₃, a homoleptic complex exhibiting isotropic EDO when doped in CBP as a reference.¹⁴⁻²³ The distribution of the emitting dipole moment of Ir(ppy)₃ was close to the random distribution with a simulated Θ value of 0.67 and random orientation of the C_3 symmetry axis of the molecule. (see Fig. S4 in Supplementary Information).

Addition: On page 11, line 237 of main text.

Meanwhile, random EDO of Ir(ppy)₃ is attributed to orthogonal intermolecular interaction sites, resulting in random orientation of the molecule.

Addition: On page 3, line 46 of Supplementary Information

Deposition simulation of Ir(ppy)₃ on CBP substrate

Deposition simulation of Ir(ppy)₃ was demonstrated on the pre-organized CBP substrate layer. Ir(ppy)₃ is a well-known homoleptic complex exhibiting isotropic emitting dipole orientation (EDO) as doped in organic host layers.^{S3-S4} The identical QM and MD methods described in the main text were employed to analyze the orientation of Ir(ppy)₃ deposited on CBP layer using Jaguar^{S5} and Desmond.^{S6} The symmetric structure of Ir(ppy)₃ results in three equivalent triplet transition dipole moments (TDMs) by ³MLCT with polar coordinates of [$\varphi_M = 87^\circ$, $\theta = 25^\circ$], [$\varphi_M = 87^\circ$, $\theta = 145^\circ$], and [$\varphi_M = 87^\circ$, $\theta = -95^\circ$], with respect to the molecular C_3 axis as shown in Figure S4a. Molecular configurations and three TDM vectors of Ir(ppy)₃ in the deposition simulation were recorded every 6 ps until reaching 6,000 ps. Figure S4b and S4c show the statistical results of the ratio of horizontal TDM (TDM_H) and the angle of the C_3 axis of Ir(ppy)₃. The distribution of TDM_H follows the random distribution line (blue line in Figure S4b) and the ensemble average of $\Theta=0.67$ is in good agreement with the EDO observed by experiments. The angular distribution of the C_3 axis is close to the random distribution line (blue line in Figure S4c) as well.

Addition: References on page 4, line 82 of Supplementary Information

S3. Liehm, P. *et al.* Comparing the emissive dipole orientation of two similar phosphorescent green emitter molecules in highly efficient organic light-emitting diodes. *Appl. Phys. Lett.* 101, 253304 (2012).

S4. Moon, C.-K., Kim, K.-H., Lee, J. W. & Kim, J.-J. Influence of host molecules on emitting dipole orientation of phosphorescent iridium complexes. *Chem. Mater.* 27, 2767–2769 (2015).

S5. Jaguar 9.2, Schrödinger, LLC, New York, NY, (2016).

S6. Desmond Molecular Dynamics System 4.6, D. E. Shaw Research, New York, NY (2016); Maestro-Desmond Interoperability Tools, Schrödinger, New York, NY (2016).

Addition: Caption of Figure S4, on page 13 of Supplementary Information

Figure S4. Quantum chemical simulation of the TDMs and molecular dynamic simulation of molecular and emitting dipole orientations of Ir(ppy)₃. (a) Three triplet TDM vectors of Ir(ppy)₃ with a 3-fold rotation symmetry from iridium to three equivalent ppy ligands by ³MLCT. The C₃ symmetry axis from the origin located at the Ir atom toward pyridines was set as **m_z**, the vector normal to the plane including **m_z** and one of Ir-N vector was set as **m_y**, and **m_x** was determined by a cross product of **m_y** and **m_z** in the dopants. Optimization of the molecular structures were performed using B3LYP method and LACVP** basis set. Spin-orbit coupled time-dependent density functional theory (SOC-TDDFT) calculations were carried out using B3LYP method and DYALL-2ZCVP_ZORA-J-PT-GEN basis set. (b) A histogram of the TDM_H of Ir(ppy)₃ with a simulated Θ value. Red bars indicate the population of the phosphor configurations having TDM_H values in steps of 0.01. The blue line is the theoretical line of TDM_H from an arbitrary vector and the green line represents the deviations between red bars and blue lines. Note that 125100 data were included in the histogram by a product of 41,700 frames and three TDMs. (c) A histogram of the angle of the C₃ axis of Ir(ppy)₃ in steps of 2°. The blue line represents an angular random distribution of an arbitrary vector. This histogram includes 41,700 data in total.

Comment 3. Conclusion

In the abstract, the authors say "Parallel alignment of the main cyclometalating ligands in the molecular complex due to host interactions rather than the ancillary ligand orienting to vacuum leads to the horizontal EDO". However, in Line 199, they also say "tmd group is less miscible in the host substrates than the main ligand ppy group, which explains the orientation of the aliphatic ancillary ligand toward vacuum side for Ir(ppy)₂tmd". The latter means that the small miscibility of the ancillary ligand also contributes to the horizontal orientation of the emission dipole moments. The authors should consider what is the principal effect and what is the secondary effect, and carefully reconstruct the logic.

Reply: The orientation could partly be interpreted by comparison of the solubility parameters between phosphors. However, we think that the solubility parameter remains controversial in dealing with a molecule at the organic/vacuum interface. Therefore, we have decided to delete the paragraph and to focus on "parallel alignment of the main cyclometalating ligands" for a clear, consistent, and compact article.

Deletion: The paragraph describing the solubility parameter was deleted in the main text.

Comment 4. Line 113

The definitions of "mx" and "my" seem opposite. In Fig. 2a, mx is normal to the molecular Ir-O-Ir plane, not in the the molecular Ir-O-Ir plane.

Comment 5. Line 116

"Fig. 2b" should be "Fig. 2a".

Reply: Thanks the reviewer for the kind comments. We have corrected them based on the reviewer's comments 4 and 5.

Comment 6. Line 136 and Fig. S3

Fig. S3 should have a proper caption (explanation) in the Supporting Information file. The meanings of the black and red curve are not explained here. (They can be understood in Fig. 5 later.)

Reply: We added the explanation of the black and red curve in the caption of Fig. S3 as follows.

Addition: Figure S3 on page 8 of Supplementary Information

Figure. S3. 50 trajectories of TDM_H up to 6000 ps in 5 combinations of the host and the dopant. Black and red lines represent the ratio of TDM_H in the range of 0 to 1 and the angle of the C₂ axis of phosphors in the range of 0° to 180°, respectively.

Comment 7. Line 139

I would like to ask the authors to give a more detailed explanation how the blue curve and red bars in Fig. 3a are calculated.

Reply: We added a more detailed explanation how the blue curve and red bars in Fig. 3a are calculated.

Revision: A caption of Figure 3 on page 22, line 428

Before –

(a) Histograms of the EDO with simulated Θ values. Green lines represent deviations of the population compared to the distribution of TDM_H of an arbitrary vector indicated as blue lines (Inset: enlarged deviation at the region of $0.8 \leq TDM_H \leq 1$) (c) Stacked histogram of the angle of the C_2 axis of phosphors. Distribution of the angle in different ranges of TDM_H is distinguished by different color.

After-

(a) Histograms of the EDO with simulated Θ values. Red bars indicate population of the phosphor configurations having TDM_H values in steps of 0.01. Blue lines are theoretical lines of TDM_H from an arbitrary vector detailed derivation of which is given in Appendix. Green lines represent deviations of the population compared to the distribution of TDM_H of an arbitrary vector. (Inset: enlarged deviation in the region of $0.8 \leq TDM_H \leq 1$) (c) Stacked histogram of the angle of the C_2 axis of phosphors with mean angles. Populations of the vector are plotted in steps of 2° . Distribution of the angle in different ranges of TDM_H is distinguished by different colors.

Comment 8. Line 178

"6 configurations" should be "5 configurations"?

Reply: Thanks for the comment. The number has been corrected to 5.

Comment 9. Line 213 and 221

It is not easy to read the energy differences from Fig. S4. I recommend that the author revise Fig. S4 and/or show the values in the main text.

Reply: Actually, Figure S4 (Figure S5 in the revised Supplementary Information) shows raw data of Figure 4b in the main text. However, the authors agree that there are lack of explanations about the figure. Therefore, we have revised the figure with increase of the scatter size and rescaling y-axis. A paragraph with the title of “*Non-bonding energies depending on the dipole orientation*” in Supplementary Information also has been revised for better understanding.

Revision: Figure S5 (Figure S4 in original manuscript is Figure S5 in the revised manuscript) on page 15 of Supplementary Information

Before – Figure S4. Relationship of the emitting dipole orientation and the non-bonding energy during the deposition.

After – Figure. S5. Relationship of the non-bonding energy of the phosphors and the emitting dipole orientation in five host-dopant systems. Red lines represent the mean non-bonding energies as a function of the ratio of TDM_H.

Revision: On page 3, line 61 of Supplementary Information

Before –

Non-bonding energies depending on the dipole orientation

Figure S4 shows the calculated non-bonding energies of the phosphors as scatters related to their dipole orientations during the deposition simulation. The energy fluctuation comes from their positions on the surface as well as thermal vibrations. Mean energies depending on the dipole orientation were used in Fig. 4b to discuss about the intermolecular interactions affecting to the molecular orientation of phosphors.

After –

Non-bonding energies for different host-dopant systems

Non-bonding energies of the phosphors located on the surface were calculated by the summation of van der Waals and Coulomb interaction energies with cut-off radius of 0.9 nm of each atom of the phosphors. Figure S5 shows the raw data of the calculated non-bonding energies of the phosphors in every frame of MD simulations (41700 scatters for a system) with their dipole orientations in the configuration. Mean non-bonding energies of the five systems are indicated as red lines, respectively, and are compared in Figure 4b in the main text to discuss the relationship between molecular orientation of phosphors and intermolecular interactions. Note that mean non-bonding energies are lowered as TDM_H increase for Ir(ppy)₂tmd:CBP, Ir(ppy)₂tmd:TSP01, Ir(3',5',4-mppy)₂tmd:TSP01, and Ir(dmppy-ph)₂tmd:CBP but there is a broad energy trap in the vertical orientation region of $TDM_H=0.1-0.5$ for Ir(ppy)₂tmd:UGH-2.

Comment 10. Line 250

"Fig. 4a" should be "Fig. S4"?

Reply: Thanks for the comment. However, in fact, Figure 4b is correct in this sentence.

Revision: On page 12, line 258

Before - Much lower (larger) non-bonding energy of Ir(dmppy-ph)₂tmd in Fig. 4a

After - Much lower (larger) non-bonding energy of Ir(dmppy-ph)₂tmd in Fig. 4b.

Reviewer #2 (Remarks to the Author):

The authors attempt to explain molecular orientations of Ir complexes in an organic host using molecular dynamics simulations. The topic is definitely interesting and timely, however the paper is rather technical and to some extent inconclusive (stating that non-bonded interactions guide the orientation does not really help). I suggest publishing in a more specialized journal.

Reply: We don't understand how the reviewer reached the conclusion that "the paper is rather technical and to some extent inconclusive (stating that non-bonded interactions guide the orientation does not really help)" even though the topic is definitely interesting.

As we described clearly in the introduction part, three different models have been proposed to explain the origin of the preferred orientation of some Ir complexes; (1) molecular aggregation of the dopants leading to randomizing their orientation by suppressing the intermolecular interaction between the dopant and host molecules, (2) strong intermolecular interactions between electro-positive sides of the dopant and the electro-negative host molecules promoting parallel alignment of the *N*-heterocycles of Ir-complexes by forming host-dopant-host pseudo-complex mainly participating in ³MLCT transition, and (3) π - π interactions between the dopant and host molecules on the organic surface bringing alignment of aliphatic ligands to the vacuum side.

Since the preferred orientation is observed in vacuum deposited films, it is very natural to perform MD simulation of the vacuum deposition process.

Our MD simulation clearly revealed that

- (1) Aggregation of phosphors is not a necessary condition for the alignment of heteroleptic Ir complexes. We performed the MD simulation by depositing single molecule on an organic substrate at a time and obtained statistics by repeating for 50 molecules deposited on different positions of the organic substrate to neglect the aggregation effect of Ir complexes. Very good consistency of the simulated EDO and experimental values clearly indicates that aggregation of phosphors is not a necessary condition for the alignment of heteroleptic Ir complexes.
- (2) Alignment of aliphatic ligands to vacuum side of heteroleptic Ir complexes is not required for preferred horizontal emitting dipole orientation (EDO). (Fig. 3) Much larger portion of Ir(ppy)₂tmd molecules align with the aliphatic ligand (tmd group in these molecules) toward the vacuum ($0^\circ < \varphi_c < 90^\circ$ in Fig. 3b) than Ir(3',5',4-mppy)₂tmd and Ir(dmppy-ph)₂tmd molecules. However, the horizontal dipole ratio of Ir(ppy)₂tmd is much lower than Ir(3',5',4-mppy)₂tmd and Ir(dmppy-ph)₂tmd. These results are the reverse direction from the prediction based on the model (3) and clearly demonstrate,

therefore, that “alignment of aliphatic ligands to the vacuum side” in the model (3) is not a necessary condition for the alignment of heteroleptic Ir complexes.

- (3) Non-bonding interaction plays pivotal role for orienting heteroleptic Ir complexes. The type and magnitude of the non-bonding interactions are different for different phosphors and hosts, leading to different EDOs. For instance, Ir(ppy)₂tmd and Ir(3',5',4-mppy)₂tmd have a quadrupole composed of two dipoles from pyridines (δ^+ charge) to Ir atom ($2\delta^-$ charge). If there is a dipole in host molecule (i.e., TSPO1), strong dipole and quadrupole interaction between $\text{P}=\text{O}^{\delta^-}$ and $\delta^+\text{H}(\text{pyridine})$ anchors one phosphor molecule to two host molecules, leading to rather horizontal orientation of pyriding-Ir-pyridine bond of the phosphors which is approximately parallel to the TDM. In contrast, if host molecule has positive surface potential [i.e., δ^+ (phenyl)₃-Si-phenyl-Si-(Phenyl)₃ δ^+], there must be repulsive force between pyridine of phosphors and host molecules so that pyridine ring must be pushed to vacuum, leading to more vertical alignment of pyriding-Ir-pyridine bond the phosphors. This interpretation was confirmed by the variation of the non-bonding energy against the orientation of the TDM.

These simulation results clearly conclude that the model (2) is the one explaining the origin of the alignment of heteroleptic Ir complexes. This conclusion is scientifically important and is very useful to design phosphors and host molecules to realize very high horizontal EDO.

For clarification of this point to readers, we have inserted two paragraphs at the beginning of Discussion section and revised other paragraphs explaining the type and magnitude of the non-bonding interaction between host and dopant affecting to alignments of the phosphors deposited on the organic semiconducting layers are added on the discussion section.

Addition: The first paragraph in Discussion section on page 9, line 179.

We performed the MD simulation by depositing each target molecule on an organic substrate at a time and obtained statistics by repeating for 50 molecules deposited on different positions of the organic substrates so that the aggregation effect of the Ir complexes is neglected. Very good consistency of the simulated EDO and experimental values clearly indicates that aggregation is not a necessary condition for the alignment of heteroleptic Ir complexes.

Alignment of aliphatic ligands of heteroleptic Ir complexes to vacuum (model 3) is not required for preferred horizontal EDO either as shown in Fig. 3. Much larger portion of the aliphatic ligand (–tmd group) of Ir(ppy)₂tmd molecules align to the vacuum side ($0^\circ < \theta < 90^\circ$ in Fig. 3b) than Ir(3',5',4-mppy)₂tmd and Ir(dmppy-ph)₂tmd molecules. However, the fraction of horizontal emitting dipole of Ir(ppy)₂tmd is much lower than Ir(3',5',4-mppy)₂tmd and Ir(dmppy-ph)₂tmd. These results are the reverse direction from the prediction based on the model and clearly demonstrate, therefore, that “alignment of aliphatic ligands to the vacuum

side” is not a necessary condition for the alignment of emitting dipole orientation in heteroleptic Ir complexes. The reason why it is not required can be understood from the following consideration.

Addition: The paragraph starting with “The type and magnitude of..” on page 11 line 226.

The type and magnitude of the non-bonding interactions are different for different phosphors and hosts, leading to different EDOs. For instance, Ir(ppy)₂tmd and Ir(3',5',4-mppy)₂tmd have a quadrupole composed of two dipoles from pyridines (δ^+ charge) to Ir atom (δ^- charge). If there is a dipole in host molecule (i.e., TSPO1), strong dipole and quadrupole interaction [$-P=O\delta^-$ and $\delta^+H(\text{pyridine})$] anchors one phosphor molecule to two host molecules, leading to rather horizontal orientation of iridium-pyridines bond of the phosphors which is approximately parallel to the TDM. In contrast, if host molecule has positive surface potential [i.e., $\delta^+(\text{phenyl})_3\text{-Si-phenyl-Si-(Phenyl)}_3\delta^+$ in UGH2], there must be repulsive force between pyridine of phosphors and host molecules so that pyridine ring must be pushed to vacuum. Dispersion force between the conjugated phenyl substituents of Ir(dmppy-ph)₂tmd and nearest neighbors anchors the pyridines onto the surface as well and lowers the energy with the molecular long axis lying on the surface.

Comment 1. The description of the MD protocol is incomplete. Was the film annealed between two deposition events? Was the rest of the molecules fixed during the annealing protocol? The supplied movie shows that there is no lateral molecular dynamics. To me this is a clear indication that the system is not equilibrated long enough between deposition events.

Reply:

(1) Was the film annealed between two deposition events? Was the rest of the molecules fixed during the annealing protocol?

Answer) No. There was no annealing between two deposition events because the deposition of the phosphor is an independent non-sequential process. After one deposition step, the organic substrate is initialized and a new phosphor is prepared at a different position above the substrate for the next deposition.

(2) The supplied movie shows that there is no lateral molecular dynamics. To me this is a clear indication that the system is not equilibrated long enough between deposition events.

Answer) The supplementary video is only an example of the deposition simulation with small lateral movement due to strong binding with host molecules on the site. Some cases of the deposition simulation show large diffusion on the surface as shown in Fig. R1. The figure exhibits the trajectory of the center of the mass of Ir(ppy)₂tmd on TSPO1 substrate in the time range of 1 ns to 6 ns. The numbers in the legend are the deposition numbers in Figure S3 in

Supplementary Information. Simulation #31 corresponds to the deposition in Supplementary Video.

Fig. R1. Trajectory of the center of the mass of Ir(ppy)₂tmd on TSP01 substrate in the time range of 1 ns to 6 ns.

(3) In term of reaching equilibrium:

We acknowledge the reviewer's point that for a single deposition event depicted in the video it does not seem to sample enough lateral degrees of freedom to cover the dopant's orientation that can result in from a wider radius of host surface that it sits on (or diffuses to). This originates from the fact that the time scale that is needed to observe the lateral degrees of freedom for a single dopant in a dynamic simulation, that is larger than the size of the dopant molecule, is much longer than that of a typical molecular dynamics simulation. This, in fact, is the very reason we sampled multiple dopant deposition process by introducing 50 independent deposition events, instead of relying upon a few single-dopant trajectories.

In addition, we attempted to sample large enough equilibrated dopant positions on host substrate starting from random initial configurations of host-dopant geometries at the surface. We took the orientation data between 1 ns to 6 ns per deposition of a molecule for the statistics because we regarded the simulation time of 1 ns is long enough to reach the thermal equilibrium. The analysis is based upon an assumption that the characteristic time to determine the orientation of dopants is in same scale of which the intermolecular interaction converges. Some of the papers simulating the vacuum deposition process granted less than hundreds of ps per deposition of a molecule, followed by annealing at the end of the deposition sequence for 5 ns at 300 K [L. Muccioli, *et al. Adv. Mater.* **23**, 4532-4536 (2011)], 10 ns at 300 K [G. Han, *et al. Adv. Mater. Interfaces* **2**, 1500329 (2015)], and 10 nanoseconds at 400 K followed by gradual reduction of temperature to 300 K [C. Tonnele, *et al. Angew.*

Chem. Int. Ed. **129**, 1-6 (2017)]. In comparison with the papers, 1 ns is a reasonable time for the equilibration in the non-sequential deposition scheme.

Another validity of our simulation in terms of reaching equilibrium comes from the result of Ir(ppy)₃. The simulation resulted in random orientation of the emitting dipole moment of Ir(ppy)₃ with a simulated Θ value of 0.67 and random distribution of the C₃ axis of the molecule (see Fig. S4 in Supplementary Information). This result corresponds to the simulation result using sequential deposition by Tonnele *et al.* and guarantees the validity of this simulation method.

The authors have stated the justification of the analysis method more explicitly in the manuscript for clarification.

Addition: On page 5, line 98

Detailed steps of preparation of the substrates are given in Fig. S1 in Supplementary Information. One of the challenges of a single-trajectory-based MD analysis for orientation during deposition is that the time scale needed to observe the entirety of lateral degrees of freedom for a single molecule is much longer than that of a typical MD simulation. As such, we introduced 50 independent deposition events per dopant, instead of relying upon a single MD trajectory for each.

Addition: On page 6, line 111

The analysis is based upon an assumption that the characteristic time to determine the orientation of dopants is in same scale of which the intermolecular interaction converges after the deposition of a dopant.

Comment 2. The authors are using a commercial MD package with the built-in OPLS force-field. This force-field was never parametrized for iridium complexes and conjugated compounds. It is therefore difficult for me to judge how trustable the simulations are.

Reply:

Contrary to the reviewer's statement, the Schrödinger commercial OPLS2005 force-field has been fit to conjugated compounds. To fit the OPLS2005 torsional parameters, a training set of 631 reference systems were used which included many aromatic systems (as described in reference 26 in the manuscript). The bonds, bends and torsions between Ir and the ligands in this work used general metal-organic parameters with equilibrium parameters set to those of the optimized structure from DFT calculations. For nonbonding interactions, the electrostatic charges on the iridium atom are based on bond charge increments obtained from electronegativity differences between the atoms bonded to the iridium atom. The Van der

Waals parameter for Ir was adopted from the Universal Force-field. The molecular orientation and interaction energies between host molecules and Ir-dopants will be dominated by the dopant-ligand:host interactions, and will be little affected by the treatment of the Ir coordination sphere. To better explain how the dopant was represented in this work, the following sentences were added to the manuscript:

“Bonds between Ir and the ligand atoms were represented by default metal-organic force constants with equilibrium parameters set to those predicted by quantum chemical calculations. The electrostatic charge on the Ir atoms were based on bond charge increments obtained from electronegativity differences between Ir and bonded ligand atom, and the Van der Waals parameter was adopted from the Universal Force Field (UFF) of Goddard and co-workers. [J. Am. Chem. Soc., 1992, 114 (25), pp 10024–10035]”

To demonstrate the validity of this treatment for organometallic systems additional calculations were carried out for two systems for which experimental data is known; the well-known Al-based electron transport material, *mer*-Alq₃ (tris(8-hydroxyquinoline)aluminum) in the amorphous phase, and *fac*-Ir(ppy)₃ (tris[2-phenylpyridinato-C2,N]iridium) the archetypal phosphorescent dopant in its crystal structure.

216 Alq₃ molecules were placed with random orientations into a periodic simulation cell and subjected to an initial 30ps NPT equilibration, followed by 40ns of NPT molecular dynamics at 300K. The same default parameters, and nonbonds scheme as described above was used with the OPLS2005 force-field for the ligands. A plot of the system density throughout the simulation is shown below:

Fig. R2. Simulated density of a Alq₃ layer.

As seen in the plot, the production run density appears to be well converged. Averaging the density over the last 20% of the production run gives a predicted density of 1.19 ± 0.003 g/cc, which is in excellent agreement with the experimental measurement by Wang et al. of 1.188 g/cc (J. Therm. Anal. Calorim. (2010) 99:117–122). To illustrate the effectiveness of the metal-organic force-field treatment for Ir compounds, the crystal structure for *fac*-Ir(ppy)₃ was subjected to 10 ns of NPT at 100K molecular dynamics. The changes in system density is shown below:

Fig. R3. Simulated density of a *fac*-Ir(ppy)₃ layer.

As seen there, the density shows only very small fluctuations away from the experimental crystal structure, indicating an accurate description of intermolecular energetics. The crystal structure density for *fac*-Ir(ppy)₃ at 100K was predicted to be 1.75 ± 0.002 g/cc, which agrees well with the experimental value of 1.787 g/cc (Eur. J. Inorg. Chem. (2010) 1613–1617). These two examples demonstrate that force-field treatment used in the submitted manuscript for Ir-dopants is sufficient to capture the dominant non-bonded interactions controlling the intermolecular interaction energetics and structure.

Comment 3. Systems size are small. I believe that one needs much better statistics to make quantitative conclusions. The error bars for averaged angles (which are very similar) are not provided.

Reply:

The size of the simulation box and the number of deposited molecules are compared with other publications summarized in the table below:

Size of simulation box	Total number of deposited molecules	reference
6.6×6.6×16.6 nm ³ (UGH-2 box)	50	this work
5.6×5.6×20.0 nm ³	224	Adv. Mater. 23, 4532-4536 (2011)
5.6×5.6×20.0 nm ³	130	Adv. Mater. 24, 3790-3798 (2014).
8.5×9.2×20.0 nm ³	400	Adv. Mater. Interfaces 2, 1500329 (2015)
8.5×8.4×20.0 nm ³	1000	Angew. Chem. Int Ed. 129, 1-6 (2017)

The size of simulation box is not small but the number of deposited molecules is small compared to other publications. One may concern that the phosphors were dispersed in the simulation and covered the surface of the substrates so duplication of data in the statistics was prevented. The figure blow shows final locations of phosphors in 50 number of depositions as

red cross marks in 5 different host-dopant systems. The border line indicates xy-planes of the UGH-2, CBP, and TSPO1 substrates with sizes of 6.6×6.6 , 5.9×6.2 , and 6.3×6.5 nm², respectively.

Fig. R4. Locations of phosphors at the end of the simulation ($t = 6$ ns).

50 independent deposition events may be considered as small for statistical analysis. However, we sampled 41,700 snapshots in the histogram by taking the configurations at each time frame of 6 ps between 1 ns to 6 ns to reduce the sampling error. Molecular configurations in all the frames are probabilities of being fixed in the film. This concept assumes that further depositions does not reorganize the orientation of the pre-deposited phosphors.

Moreover, we have carried out additional 50 deposition simulations of Ir(ppy)₂tmd onto the TSPO1 substrate with different initial phosphor positions to prove consistency of the result with extended time. The new simulation resulted in Θ and mean angle of the C_2 axis (φ_c) of 0.73 and 70°, respectively, as shown in the figure below. The little differences of Θ and φ_c between the values in the manuscript and newly simulated ($\Delta\Theta=0.01$ and $\Delta\varphi_c=1^\circ$) verify reliability of the deposition simulation using 50 number of dopant molecules. Based on the result, we have added error bars of ± 0.01 to EDO and $\pm 1^\circ$ to the mean angle in Figure 3.

Fig. R5. Simulated emitting dipole orientation (EDO) and angle of the C_2 axis of Ir(ppy)₂tmd deposited on the TSPO1 substrate with new 50 initial locations of Ir(ppy)₂tmd molecules.

Revision: Figure 3 on page 26 has been revised to include the error bars.

Comment 4. I do not think that the solubility parameters are useful to predict molecular orientations. We are dealing with the organic-vacuum interface: The processes here are very different from the bulk (2D diffusion, interfacial dipole moments, etc). I do not see a physical reason why solubility should correlate to molecular orientations.

Reply: We appreciate the reviewer for the comment. We understand the reviewer’s argument that deposition and solvation are two fundamentally different processes. The orientation could partly be interpreted by comparison of the solubility parameters between phosphors and functional groups in a molecule. However, we think that the solubility parameter remains controversial in dealing with a molecule at the organic/vacuum interface. Therefore, we have decided to delete the paragraph and to focus on “parallel alignment of the main cyclometalating ligands” for a clear, consistent, and compact article.

Deletion: We deleted the paragraph describing the solubility from the manuscript.

Reviewer #3 (Remarks to the Author):

This work mainly describes the orientation of phosphors like Ir-complexes and its influences on an enhancement of outcoupling efficiency of light. What I concerned is the following two points:

Comment 1. The crystal structure of the first Ir-complex [Ir(ppy)₂(tmd) containing water was reported. Please the authors to address what the orientation in crystalline [Ir(ppy)₂(tmd) as phosphors in in-plane and out of plane. Also add their XRD patterns (in- and out-plane) into text to support your claims.

Reply:

This manuscript deals with doping systems and the phosphor does not form a crystal as doped in the organic amorphous layers if doping concentration is low. The directional host-dopant intermolecular interactions on the surface induce the phosphor orientation even though they do not form crystals. The authors respect the reviewer's comments but crystal structures of Ir(ppy)₂tmd, Ir(3',5',4-mppy)₂tmd, and Ir(dmppy-ph)₂tmd are beyond the scope of this manuscript.

Comment 2. In introductions, the molecular orientation of thin film is essential in physical properties especially extremely anisotropic properties: dielectric, ferroelectric. So, the authors should cite some of works relative to this field.

Reply: Thanks for the comment. We have added two citations include the influence on dielectric and ferroelectric properties.

Revision: On page 2, line 24

Before – Orientation of molecules in molecular films dictates their electrical and optical properties such as charge mobility,¹⁻² birefringence,³ absorption,⁴ emission,⁵ ionization potential.⁶

After – Orientation of molecules in molecular films dictates their electrical and optical properties such as charge mobility,¹⁻² birefringence,³ absorption,⁴ emission,⁵ ionization potential,⁶ and dielectric⁷ and ferroelectric properties.⁸

Addition: References on page 16, line 339.

7. Yamao, T., Yamamoto, K., Taniguchi, Y., Miki, T. & Hotta, S. Laser oscillation in a highly anisotropic organic crystal with a refractive index of 4.0, *J. Appl. Phys.* **103**, 093115 (2008).

8. Horiuchi, S. & Tokura, Y. Organic ferroelectrics, *Nature Mater.* **7**, 357-366 (2008).

Comment 3. I encouraged the authors to determine three Ir-complex crystal structures. After done, this is very nice paper.

Reply: As mentioned at the reply of comment 1 by the same reviewer, the phosphors do not form crystals as doped in organic semiconducting layers. The authors are sorry to the reviewer but determination of the crystal structures requires lot of work and is not necessary in this manuscript.

Detailed response to the reviewers' comments

Title: Unraveling the orientation of phosphors doped in organic semiconducting layers

Author: Jang-Joo Kim; Seoul National University

We appreciate the reviewers' valuable comments and we revised the manuscript. Followings are the detailed responses to the reviewers' comments.

Reviewer #1 (Remarks to the Author):

Comment: I think that the revisions by the authors were made appropriately. However, regarding Comment #3, the authors completely deleted one long paragraph in their discussions, which is essentially related to the orientation mechanism. They only deleted a paragraph and did not add any explanation for understanding the mechanism. I think that they should clearly explain the mechanism of "parallel alignment of the main cyclometalating ligands" in more detail to support their conclusion.

Reply: We concluded that the discussion about the solubility parameter comparing the compatibility of the ancillary ligand and the main cyclometalating ligands to the substrate is not essential and may distract the main point because the orientation of the ancillary ligand is not closely related to the emitting dipole orientation of phosphors. Instead of the complete deletion of the paragraph, we added a short paragraph in the discussion section of the revised manuscript as follows

“Consideration of the solubility parameters of the molecules used in the simulation also supports the simulation results as described in Supplementary Information in detail. The differences in the solubility parameters between the host and the phosphor are much less than 7 MPa^{1/2}, suggesting that all the phosphors are miscible with the hosts. In addition to that, Ir(3',5',4-mppy)₂tmd and Ir(dmppy-ph)₂tmd are less miscible than Ir(ppy)₂tmd in TSPO1. Since the difference in the miscibilities comes from the difference in main ligands of the Ir complexes, the result indicates that both 3',5',4-mppy and dmppy-ph groups are less miscible with the host than ppy of Ir(ppy)₂tmd and less preference to attachment to the substrate compared to ppy group. Therefore, the orientations of the ancillary ligand of the two phosphors [Ir(3',5',4-mppy)₂tmd and Ir(dmppy-ph)₂tmd] are more randomized during the deposition but with more horizontal orientation of emitting dipoles than Ir(ppy)₂tmd, consistent with the simulated distributions in Fig. 3b.”

We also added a paragraph in Supplementary Information as follows:

“The distributions of the ancillary ligand can be partly explained by analyzing the differences in Hildebrand solubility parameters (δ) of the phosphor and host molecules. Predicted solubility parameters of the molecules are 16.2 (UGH-2), 18.5 (CBP), 17.2 (TSPO1), 15.3 (Ir(ppy)₃), 14.7 (Ir(ppy)₂tmd), 13.8 (Ir(3',5',4-mppy)₂tmd) and 14.4 MPa^{1/2} (Ir(dmppy-ph)₂tmd), respectively, calculated from OPLS_2005 NPT MD by the equation:^{S7}

$$\delta = \left(\frac{\Delta E_v}{V_m} \right)^{1/2} \quad (1)$$

where ΔE_v is the internal energy change of vaporization, and V_m is the molar volume, respectively. In general, the differences in the solubility parameters ($\Delta\delta$) between two components in a chemical mixture can be an indicator of the degree of miscibility, with smaller and larger values of $\Delta\delta$ indicating more and less miscible, respectively. In this work, the host and the phosphor $\Delta\delta$ are much less than 7 MPa^{1/2}, suggesting that all the phosphors are miscible with the hosts. However, $\Delta\delta$'s between Ir(ppy)₃ and the hosts are smaller than between Ir(ppy)₂tmd and the hosts.^{S8} Since the difference comes from the ppy and tmd groups, it follows that the tmd group is less miscible in the host substrates than the main ligand ppy group, which explains the orientation of the aliphatic ancillary ligand toward vacuum side for Ir(ppy)₂tmd. On the other hand, the difference in the solubility parameters among Ir(ppy)₂tmd, Ir(3',5',4-mppy)₂tmd and Ir(dmppy-ph)₂tmd comes from the difference in main ligands. The reduced solubility of Ir(3',5',4-mppy)₂tmd and Ir(dmppy-ph)₂tmd indicates that both 3',5',4-mppy and dmppy-ph groups are less miscible to the host than ppy of Ir(ppy)₂tmd and less preference to attachment to the substrate compared to ppy group. Therefore, the orientations of the ancillary ligand of the two phosphors are more randomized during the deposition, consistent with the simulated distributions in Fig. 3b.”

Reviewer #3 (Remarks to the Author):

Comment: The authors have addressed the concerns raised in the previous round of review, and I agree their explanation. The work is important and the results are convincing. Therefore, the paper is now appropriate for publication.

Reply: We appreciate the reviewer’s positive comments and advice.

Reviewer #4 (Remarks to the Author):

Comment 1:

I agree with reviewer 2 that the number of events sampled is very small and it is unclear whether the sampling of the trajectory actually helps, because the authors state that some molecules remain fixed (i.e. they contribute always the same value across the whole trajectory), while other molecules change conformation. In order to quantify this, it would be

nice to compute the autocorrelation time of the angle and plot a histogram of this for the 50 molecules. This should tell us the average number of independent conformations during the trajectory.

Reply: We estimated the integrated autocorrelation times and the effective number of independent samples to validate if the 50-independent-deposition events are sufficient to represent configurations of Ir compounds doped in organic semiconducting layers. The autocorrelation coefficient (τ) as a function of the time lag (t) was calculated by

$$\tau = \frac{1}{50} \sum_{k=1}^{50} \frac{C_A(t, k)}{C_A(0, k)},$$

where k is the simulation number among the 50 deposition events and C_A is the covariance following $C_A(t) = \langle (A(s+t) - \bar{A})(A(s) - \bar{A}) \rangle$. Figure S6 exhibits τ 's of the 5 different deposition systems with the x-axis as a product of the lag and the time step of 6 ps. The integrated autocorrelation times (τ_{int}) and the effective number of independent samples (n_{eff}) were calculated using the following formulas:

$$\tau_{\text{int}} = 6 \text{ ps} \times \sum_{t=0}^{t_{\text{final}}} \tau(t),$$

$$n_{\text{eff}} = 50 \times \frac{5000}{2\tau_{\text{int}}}.$$

τ_{int} values were 157, 168, 189, 218, and 208 ps, and n_{eff} values were 765, 715, 638, 557, and 583 for UGH-2:Ir(ppy)₂tmd, CBP:Ir(ppy)₂tmd, TSPO1:Ir(ppy)₂tmd, TSPO1:Ir(3',5',4-mppy)₂tmd, and TSPO1:Ir(dmppy-ph)₂tmd systems, respectively. The small τ_{int} values compared to the total simulation time of 5000 ps (1000 ps to 6000 ps) indicate no large autocorrelation in the time series data. We concluded that the number of samples is sufficient for the analysis of molecular configuration in the trajectories.

Figure R1. Autocorrelation coefficients in the 5 different deposition systems.

We added a following sentence (line 181-186) in the revised manuscript. In addition to that, Figure S6 and its explanation were added in Supplementary Information.

“The computation of the autocorrelation times of the molecular angles for the 50 independent depositions for each phosphorescent molecule verifies that the number of events sampled during the deposition (the simulation time and the number of the deposition events) is large enough to validate the MD simulation of the vacuum deposition process. (Fig. S6 in Supplementary Information)”

Comment 2:

On the positive side, the agreement between the experimental and the theoretical values is amazingly good.

Reply: Thanks for the positive comment.

Comment 3:

I think the manuscript in the present form is lacking a message beyond the fact that the protocol seems to work. I concur with the reviewer on point 1 that stating that “nonbonded interactions” are responsible in a system where no new bonds are formed or broken is not really illuminating. In their response, the authors claim that they can distinguish aliphatic from electrostatic mechanisms in their reply to the reviewer and this could be easily demonstrated by comparing the relative contribution of the nonbonded contributions of the vdW energy and electrostatic energy as a function of orientation which their program can just print out. As a function of orientation we should have an energy profile for both contributions to the nonbonded energy and if the authors are correct, the variation of the vdW energy should be weaker than that of the Coulomb energy.

Reply: We proposed the local intermolecular interactions between aromatic ligands of the Ir compounds and nearest host molecules on the surface as a factor determining the molecular orientation of the phosphor in the paper. The proposed local interactions were dipole(quadrupole)-dipole, dipole(quadrupole)-induced dipole, and induced dipole-induced dipole interactions. In the molecular dynamics simulation, the dipole(quadrupole)-dipole interactions contribute to Coulomb energy and dipole(quadrupole)-induced dipole along with induced dipole-induced dipole interactions are included in vdW energy. Therefore, reduction of vdW energy by the molecular alignments is in accordance with our explanation.

The separated vdW and Coulomb energies as a function of the molecular orientation are depicted in Figure R2. The vdW energies in CBP:Ir(ppy)₂tmd, TSP01:Ir(ppy)₂tmd, TSP01:Ir(3',5',4-mppy)₂tmd, and TSP01:Ir(dmppy-ph)₂tmd systems decrease whereas the energy in UGH-2:Ir(ppy)₂tmd increases as the ratio of the horizontal transition dipole

moment (TDM_H) increases. For a polar host molecule of TSPO1, Coulomb energies of the phosphors are lowered as TDM_H increases. The variation of vdW energy depending on the molecular orientation was 3–5 kcal/mol which is larger than the variation of Coulomb energy of 0-1 kcal/mol. The results indicate that vdW interactions (dipole-induced dipole and induced dipole-induced dipole interactions) between the aromatic ligands and the nearest host molecules are the main mechanism contributing to the molecular alignment of the phosphors. The Coulomb interaction helps further alignments of the phosphors if polar host materials are employed.

We have revised the manuscript to explain contributions of dispersion and electrostatic interactions to the molecular alignment.

Figure R2. VdW and Coulomb interaction energies as a function of the emitting dipole orientation.

Revision: line 246 on page 12

Before:

The type and magnitude of the non-bonding interactions are different for different phosphors and hosts, leading to different EDOs. For instance, Ir(ppy)₂:tmd and Ir(3',5',4-mppy)₂:tmd have a quadrupole composed of two dipoles from pyridines (δ^+ charge) to Ir atom ($2\delta^-$ charge). If there is a dipole in host molecule (i.e., TSPO1), strong dipole and quadrupole interaction [$-P=O^{\delta^-}$ and $\delta^+H(\text{pyridine})$] anchors one phosphor molecule to two host molecules, leading to rather horizontal orientation of iridium-pyridines bond of the phosphors which is approximately parallel to the TDM.

After:

The type and magnitude of the non-bonding interactions are different for different phosphors and hosts, leading to different EDOs as shown in Fig. S8 in Supplementary Information. Dispersion interaction between the aromatic ligands and nearest host molecules contributes to the molecular alignment of the phosphors and electrostatic interaction between them helps the further alignments. For instance, Ir(ppy)₂:tmd and Ir(3',5',4-mppy)₂:tmd have a quadrupole composed of two dipoles from pyridines (δ^+ charge) to Ir atom ($2\delta^-$ charge). If there is a

dipole in host molecule (i.e., TSPO1), dipole and quadrupole interaction [$-\text{P}=\text{O}^{\delta-}$ and $\delta^+\text{H}(\text{pyridine})$] anchors one phosphor molecule to two host molecules, leading to rather horizontal orientation of iridium-pyridines bond of the phosphors which is approximately parallel to the TDM.

Comment 4:

Going beyond the comments of reviewer 2 in this respect, my worry is that the aliphatic interaction with a substrate of just 256 molecules is too weak. The interaction falls like r^{*-6} with distance, but integrated over the bulk, this makes a huge contribution. Therefore, the contribution of the vdW interactions may be underestimated by the present model of the substrate. The authors could repeat the 50 simulations for one system with 1024 molecules as a substrate and demonstrate that the variation in the vdW energy does not change significantly.

Reply: The vdW and Coulomb interaction energies were calculated with a cut-off radius of 0.9 nm of each atom of the phosphors. Therefore, use of more host molecules in the substrates does not result in larger vdW energies both by aliphatic and aromatic interactions. The sizes of the substrate surface were 6.6×6.6 , 5.9×6.2 , and 6.3×6.5 nm² for the UGH-2, CBP, and TSPO1 substrates which are large enough to avoid self-correlation in the intermolecular interactions.

In addition, we performed the vacuum deposition simulation of Ir(ppy)₂tmd on a TSPO1 substrate consisting of 1024 molecules. The volume of the substrate was $10.2 \times 10.5 \times 8.2$ nm³ which are about 4-fold larger than the substrate consisting of 256 molecules as shown in Figure R3a. Histograms of the EDO and angle of the C₂ axis are plotted in Figure R3b and R3c. The simulated EDO value (Θ , the ratio of horizontal transition dipole moment) was 0.73, very close to the EDO value of 0.74 in the main text simulated using the substrate consisting of 256 molecules. The distribution of C₂ axis was also similar to the distribution in the main text with the rather random orientation but a little preferred orientation to the vacuum side. The results indicate that the deposition simulation with the 256-molecule substrate estimate the vdW interaction adequately and is rather cost-effective compared to simulation with a large number of molecules.

We have added this results to Figure S5 in Supplementary Information.

Figure R3. (a) Size of the TSPO1 substrates containing 256 and 1024 molecules. Histograms of (b) TDM_H and (c) the angle of the C_2 axis by 50 deposition events of $Ir(ppy)_2tmd$ on the 1024-molecule-TSPO1 substrate.

Addition: line 179 on page 9

The size of the substrates turns out to be large enough to simulate the vacuum deposition of the phosphorescent dyes adequately, as confirmed by the similar results obtained on a larger substrate consisting of 1024 molecules (Fig.S5 in Supplementary Information).

Addition: line 61 on page 3 of Supplementary Information

Deposition simulation of $Ir(ppy)_2tmd$ on a 1024-molecule-TSPO1 substrate

To validate if the substrates consisting of 256 molecules appropriately consider intermolecular interactions, the deposition simulation with a larger number of host molecules was performed. Figure S5a exhibits the equilibrated 1024-molecule-TSPO1 substrate. Figure S5b and S5c show histograms of TDM_H and angle of the C_2 axis of $Ir(ppy)_2tmd$, respectively, deposited on the 1024-molecule substrate. The simulated EDO value (Θ , the ratio of horizontal transition dipole moment) was 0.73 which is very similar to the value of 0.74 obtained from the simulation using the substrate consisting of 256 TSPO1 molecules. The distribution of the C_2 axis vector based on the 1024-molecule substrate was broad with a mean angle of 74°, exhibiting little difference from the molecular distribution based on the substrate consisting of 256-molecule.

Addition: Figure S5 on page 15 of Supplementary Information

Figure S5. (a) Comparison of TSPO1 substrates consisting of 256 and 1024 molecules. (b) Histograms of (a) TDM_H and (b) φ_C demonstrated by the deposition simulation of Ir(ppy)₂tmd on the 1024-molecule TSPO1 substrate